# Discovering Latent Knowledge in Language Models Without Supervision

**Collin Burns**[*]
UC Berkeley

**Haotian Ye**[*]
Peking University

**Dan Klein**
UC Berkeley

**Jacob Steinhardt**
UC Berkeley

## ABSTRACT

Existing techniques for training language models can be misaligned with the truth: if we train models with imitation learning, they may reproduce errors that humans make; if we train them to generate text that humans rate highly, they may output errors that human evaluators can't detect. We propose circumventing this issue by directly finding latent knowledge inside the internal activations of a language model in a purely unsupervised way. Specifically, we introduce a method for accurately answering yes-no questions given only unlabeled model activations. It works by finding a direction in activation space that satisfies logical consistency properties, such as that a statement and its negation have opposite truth values. We show that despite using no supervision and no model outputs, our method can recover diverse knowledge represented in large language models: across 6 models and 10 question-answering datasets, it outperforms zero-shot accuracy by 4% on average. We also find that it cuts prompt sensitivity in half and continues to maintain high accuracy even when models are prompted to generate incorrect answers. Our results provide an initial step toward discovering what language models know, distinct from what they say, even when we don't have access to explicit ground truth labels.

## 1 INTRODUCTION

The increasing deployment of language models in real-world applications opens up exciting possibilities, but it also raises the stakes of AI research and presents new risks (Bommasani et al., 2021; Weidinger et al., 2021; Bender et al., 2021). One of these risks is that language models do not always output text that is true (Evans et al., 2021; Hendrycks et al., 2021; Kenton et al., 2021).

Common training objectives can cause models to learn internal representations related to truth, since truth is a useful feature for many tasks. However, these objectives can also cause language models to output text that is false, at least in some circumstances. For example, if we train a model to imitate human-generated text, it may learn to output common misconceptions (Lin et al., 2022). Or if we train a chat bot to optimize a reward such as engagement, it may learn to generate text that is compelling but false (Roller et al., 2021). If we try to reward model outputs that look true, a model may still learn to output false text if human raters can't evaluate the correctness of that text (Kenton et al., 2021).

In each case, this is an issue that stems from the misalignment between a training objective and the truth. As models are applied to more complex domains, human supervision may become less effective at mitigating this misalignment. Moreover, because this is a problem with the training objective rather than a model's capabilities, it likely won't be solved by scaling up models alone.

We propose a different approach for addressing this misalignment: using models to answer questions in a purely *unsupervised* way. Intuitively, instead of trying to explicitly, externally specify truth, we search for implicit, internal "beliefs" or "knowledge" learned by a model. We approach this problem by leveraging the fact that a model's representation of truth must satisfy logical consistency properties, which are unlikely to be satisfied by many other features.

We implement this idea by introducing Contrast-Consistent Search (CCS), a method that learns a linear projection of the hidden states that is consistent across negations, as illustrated in Figure 1. We find that despite its simplicity and despite not having access to any labels or model outputs,

---

[*]Equal contribution.

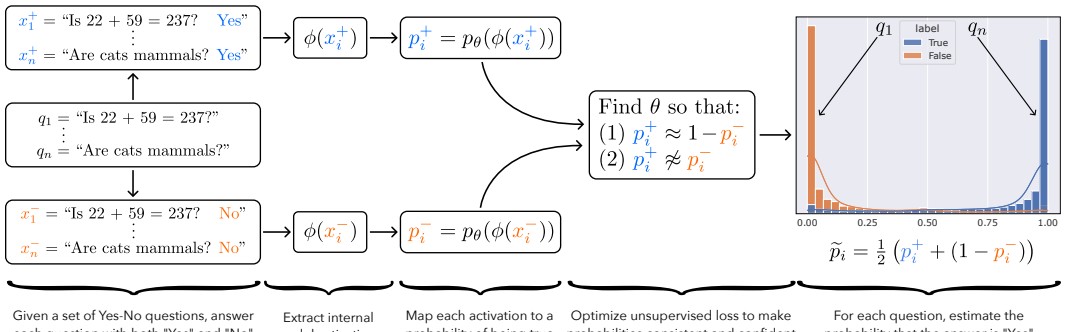

Figure 1: An illustration of our method, Contrast-Consistent Search (CCS). For each yes-no question $q_i$, we let $x_i^+$ and $x_i^-$ be the natural language statements where we answer $q_i$ as "Yes" and "No" respectively. Answering the question $q_i$ then amounts to determining which of $x_i^+$ or $x_i^-$ is true. We compute probabilities $p_i^+$ and $p_i^-$ that $x_i^+$ and $x_i^-$ are true respectively using a learned mapping from the hidden states to a number between $0$ and $1$. We search for a mapping such that that the probabilities are both confident and consistent. On the right, we show a histogram of the "Yes" probabilities, $\tilde{p}_i = 0.5 \cdot (p_i^+ + (1 - p_i^-))$, learned by our method on the unlabeled train split of the COPA dataset (Roemmele et al., 2011) with the UnifiedQA model (Khashabi et al., 2020). Our method uses no labels and no model outputs, but still learns to accurately answers questions.

CCS can accurately recover knowledge from model representations: evaluated across 6 models and 10 question-answering datasets, CCS outperforms the accuracy of strong zero-shot baselines by 4% on average (Section 3.2.1). The resulting classifier is also less sensitive to different prompts than zero-shot, cutting the standard deviation in accuracy in half. Additionally, we try deliberately prompting models to make incorrect outputs, which should intuitively change what models say but which shouldn't affect their latent knowledge. We find that this causes zero-shot accuracy to drop by up to 9.5% (Section 3.2.2) without decreasing the accuracy of CCS.

We systematically analyze CCS to understand the features it discovers. We show that it transfers across unrelated tasks, suggesting that models may have a task-agnostic representation of the truth and that CCS is able to approximately discover it (Section 3.3.1). Moreover, CCS sometimes works best using the hidden states in the *middle* layers of a network and can work even when model outputs aren't very informative, suggesting that it can leverage different features from those used by the outputs (Section 3.3.2). Finally, we show that representations of truth tend to be salient in models: they can often be found without much data, and they can often be found by taking the top principal component of a slightly modified representation space (Section 3.3.3).

Most existing techniques for making models truthful use human supervision to explicitly specify what is correct. However, it is not feasible to provide supervision in some settings. Our work suggests that an external source of ground truth may not actually be necessary: we may instead be able to find a model's latent representation of truth, independent of what a model says, without using any supervision in the first place.

## 2    PROBLEM STATEMENT AND FRAMEWORK

In this section we describe our problem setup in more detail and introduce Contrast-Consistent Search (CCS), a method for discovering latent knowledge in language models without supervision.

### 2.1    PROBLEM: DISCOVERING LATENT KNOWLEDGE

Given a pre-trained neural language model and a set $q_1, \ldots, q_n$ of yes-no questions[1], our goal is to answer each $q_i$ correctly. Here, $q_i$ can be any question with a well-defined answer, including procedural questions like "Is 22+59 = 237?", for which the answer is "No", and factual questions like "Are cats mammals?", for which the answer is "Yes".

---

[1]Technically, we only require that there are two mutually exclusive answers. For example, we can also use the labels "positive" and "negative" for sentiment classification. Moreover, our setup can easily extend to the case where we want to evaluate the truth of a set of statements instead of answering a set of questions.

Importantly, we want methods that do not rely on the model generating correct outputs and that do not rely on external supervision. Instead, we turn to the model's unlabeled hidden representations. Specifically, let $\phi(x) \in \mathbb{R}^d$ denote some feature representation on a natural language input $x$, such as the hidden states of a Transformer-based language model. Our goal is to answer the questions $q_1, \ldots, q_n$ only given access to $\phi(\cdot)$. In Section 2.2 we introduce a method for this problem that attains high accuracy (Section 3), demonstrating that this task is tractable.

## 2.2 METHOD: CONTRAST-CONSISTENT SEARCH

To make progress on the goal described above, we exploit the fact that truth has special structure: it satisfies consistency properties that few other features in a language model are likely to satisfy. Our method, Contrast-Consistent Search (CCS), leverages this idea by finding a direction in activation space that is consistent across negations. As we illustrate in Figure 1, CCS works by (1) answering each question $q_i$ as both "Yes" ($x_i^+$) and "No" ($x_i^-$), (2) computing the representations $\phi(x_i^+)$ and $\phi(x_i^-)$ of each answer, (3) mapping the answer representations to probabilities $p_i^+$ and $p_i^-$ of being true, then (4) optimizing that mapping so that the probabilities are both consistent and confident.

Concretely, the input to CCS is a set of Yes-No questions, $q_1, \ldots, q_n$, and access to a pretrained model's representations, $\phi(\cdot)$; the output of CCS is a lightweight probe on top of $\phi(\cdot)$ that can answer new questions. Here, $\phi(\cdot)$ is fixed but should contain useful information about the answers to $q_1, \ldots, q_n$, in the sense that if one *did* (hypothetically) have access to the ground-truth labels for $q_1, \ldots, q_n$, one would be able to train a small supervised probe on $\phi(\cdot)$ that attains high accuracy. Importantly, CCS does not modify the weights of the pretrained model and it does not use labels.

**Constructing contrast pairs.** An important property that truth satisfies is negation consistency: the answer to a clear-cut question cannot be both "Yes" and "No" at the same time, as these are negations of each other. Probabilistically, for each question $q_i$, the probability that the answer to $q_i$ is "Yes" should be one minus the probability that the answer to $q_i$ is "No". To use this property, we begin by constructing contrast pairs: for each question $q_i$, we answer $q_i$ both as "Yes", resulting in the new natural language statement $x_i^+$, and as "No", resulting in the natural language statement $x_i^-$. We illustrate this in Figure 1 (left). We will then learn to classify $x_i^+$ and $x_i^-$ as true or false; if $x_i^+$ is true, then the answer to $q_i$ should be "Yes", and if $x_i^-$ is true, then the answer to $q_i$ should be "No".

In practice, we convert each task into a question-answering task with two possible labels, then we use task-specific zero-shot prompts to format questions and answers as strings to construct each contrast pair. The opposite labels we use to construct contrast pairs can be "Yes" and "No" for a generic task, or they can be other tasks-specific labels, such as "Positive" and "Negative" in the case of sentiment classification. We describe the exact prompts we use to for each task in Appendix B.

**Feature extraction and normalization.** Given a contrast pair $(x_i^+, x_i^-)$, CCS first computes the representations $\phi(x_i^+)$ and $\phi(x_i^-)$ using the feature extractor $\phi(\cdot)$. Intuitively, there are two salient differences between $\phi(x_i^+)$ and $\phi(x_i^-)$: (1) $x_i^+$ ends with "Yes" while $x_i^-$ ends with "No", and (2) one of $x_i^+$ or $x_i^-$ is true while the other is false. We want to find (2) rather than (1), so we first try to remove the effect of (1) by normalizing $\{\phi(x_i^+)\}$ and $\{\phi(x_i^-)\}$ independently. In particular, we construct normalized representations $\tilde{\phi}(x)$ as follows:

$$\tilde{\phi}(x_i^+) := \phi(x_i^+) - \mu^+ ; \quad \tilde{\phi}(x_i^-) := \phi(x_i^-) - \mu^- ,$$

where $\mu^+, \mu^- \in \mathbb{R}^d$ are the means of $\{\phi(x_i^+)\}_{i=1}^n$ and $\{\phi(x_i^-)\}_{i=1}^n$. This normalization ensures that $\{\tilde{\phi}(x_i^+)\}$ and $\{\tilde{\phi}(x_i^-)\}$ no longer form two separate clusters. In practice we also normalize the scale of the features, but this isn't essential for the method to work; see Appendix G.1 for details.

**Mapping activations to probabilities.** Next, we learn a probe $p_{\theta,b}(\tilde{\phi})$ that maps a (normalized) hidden state $\tilde{\phi}(x)$ to a number between 0 and 1 representing the probability that the statement $x$ is true. We use a linear projection followed by a sigmoid $\sigma(\cdot)$, i.e. $p_{\theta,b}(\tilde{\phi}) = \sigma(\theta^T \tilde{\phi} + b)$, but nonlinear projections can also work. For simplicity, we sometimes omit the $\theta, b$ subscript in $p$.

**Training objective.** To find features that represent the truth, we leverage the consistency structure of truth. First, we use the fact that a statement and its negation should have probabilities that add up to 1. This motivates the consistency loss:

$$L_{\text{consistency}}(\theta, b; q_i) := \left[ p_{\theta,b}(x_i^+) - (1 - p_{\theta,b}(x_i^-)) \right]^2$$

However, this objective alone has a degenerate solution: $p(x^+) = p(x^-) = 0.5$. To avoid this problem, we encourage the model to also be confident with the following confidence loss:

$$L_{\text{confidence}}(\theta, b; q_i) := \min\{p_{\theta,b}(x_i^+), p_{\theta,b}(x_i^-)\}^2$$

We can equivalently interpret $L_{\text{confidence}}$ as imposing a second consistency property on the probabilities: the law of excluded middle (every statement must be either true or false). The final unsupervised loss is the sum of these two losses, averaged across all contrast pairs:

$$L_{\text{CCS}}(\theta, b) := \frac{1}{n} \sum_{i=1}^{n} L_{\text{consistency}}(\theta, b; q_i) + L_{\text{confidence}}(\theta, b; q_i)$$

Note that both losses are necessary; $L_{\text{confidence}}$ alone also has a degenerate solution.

**Inference.** Both $p(x_i^+)$ and $1 - p(x_i^-)$ should represent the probability that the answer to $q_i$ is "Yes". However, because we use a soft consistency constraint, these may not be exactly equal. To make a prediction on an example $x_i$ after training, we consequently take the average of these:

$$\tilde{p}(q_i) := \frac{1}{2}(p(x_i^+) + (1 - p(x_i^-)))$$

We then predict that the answer to $q_i$ is "Yes" based on whether $\tilde{p}(q_i)$ is greater than 0.5. Technically, we also need to determine whether $\tilde{p}(q_i) > 0.5$ corresponds to "Yes" or "No," as this isn't specified by $L_{\text{CCS}}$. For simplicity in our evaluations we take the maximum accuracy over the two possible ways of labeling the predictions of a given test set. However, in Appendix A we describe how one can identify the two clusters without any supervision in principle by leveraging conjunctions.

## 3 RESULTS

### 3.1 EXPERIMENTAL SETUP

Here we give an overview of our experimental setup; see Appendix G for full details. We provide code at https://www.github.com/collin-burns/discovering_latent_knowledge.

**Models.** We test six models: encoder-decoder models (T5 (Raffel et al., 2020), UnifiedQA (Khashabi et al., 2020), T0 (Sanh et al., 2021)), autoregressive models (GPT-J (Wang & Komatsuzaki, 2021)), and encoder-only models (RoBERTa (Liu et al., 2019), DeBERTa (He et al., 2021)).

**Data.** We test models on 10 datasets: sentiment classification (IMDB (Maas et al., 2011) and Amazon (McAuley & Leskovec, 2013)), topic classification (AG-News (Zhang et al., 2015) and DBpedia-14 (Lehmann et al., 2015)), NLI (RTE (Wang et al., 2018) and QNLI (Rajpurkar et al., 2016)), story completion (COPA (Roemmele et al., 2011) and Story-Cloze (Mostafazadeh et al., 2017)), question answering (BoolQ (Clark et al., 2019)), and common sense reasoning (PIQA (Bisk et al., 2020)).

We convert each dataset to a yes-no question-answering task or a binary classification task, as described in Appendix G. We balance the labels and randomly subsample 1000 examples from each dataset (except for COPA, which has only 500 examples total), then randomly split each dataset into an unsupervised training set (60% of the data) and test set (40%). We subsample each dataset for computational efficiency reasons; because we aggregate over 9 prompts per dataset, 10 datasets, and 6 models, 1000 datapoints per dataset actually corresponds to approximately 180k examples in total.

**Methods.** We test four main methods: zero-shot, calibrated zero-shot, Contrast-Consistent Search (CCS), and Logistic Regression (LR). Zero-shot works by predicting the answer with the highest log probability according to the language model, averaged across the tokens that make up that label. Calibrated zero-shot works by balancing zero-shot predictions to be $50/50$ for each answer, as we describe in more detail below, similar to Zhao et al. (2021). For Logistic Regression we train on the training split for each dataset using $(\tilde{\phi}(x^+), \tilde{\phi}(x^-))$ as the covariates, then evaluate on the corresponding test split. We treat LR as a ceiling since it uses labeled data.

When testing CCS, we optimize it 10 times using AdamW (Loshchilov & Hutter, 2017) with learning rate 0.01, then take the run with the lowest unsupervised loss. Unless otherwise specified, we train CCS using all prompts for a single training set, then evaluate it on the corresponding test split.

**Zero-shot baselines.** Zero-shot outputs sometimes suffer from miscalibration (Zhao et al., 2021), in which models are biased towards predicting specific answers. Calibrating the outputs to be uniform

| Method | RoBERTa | DeBERTa | GPT-J | T5 | UQA | T0* | Mean* |
|---|---|---|---|---|---|---|---|
| 0-shot | 60.1(5.7) | 68.6(8.2) | 53.2(5.2) | 55.4(5.7) | 76.8(9.6) | 87.9(4.8) | 62.8(6.9) |
| Calibrated 0-shot | **64.3(6.2)** | 76.3(6.0) | 56.0(5.2) | 58.8(6.1) | 80.4(7.1) | 90.5(2.7) | 67.2(6.1) |
| CCS | 62.1(4.1) | **78.5(3.8)** | 61.7(2.5) | 71.5(3.0) | 82.1(2.7) | 77.6(3.3) | 71.2(3.2) |
| CCS (All Data) | 60.1(3.7) | 77.1(4.1) | **62.1(2.3)** | **72.7(6.0)** | **84.8(2.6)** | 84.8(3.7) | **71.5(3.7)** |
| LR (Ceiling) | 79.8(2.5) | 86.1(2.2) | 78.0(2.3) | 84.6(3.1) | 89.8(1.9) | 90.7(2.1) | 83.7(2.4) |

Table 1: Accuracy of each method and model averaged across all prompts and dataset, with the average standard deviation of accuracy across different prompts shown in parentheses. For most models, CCS outperforms zero-shot accuracy and exhibits lower sensitivity to prompts, even though this was not our goal. This shows that we can recover knowledge from language model activations without supervision, and can do so in a way that is competitive with strong baseline methods that use model outputs. *T0 was trained on 9 out of 10 of the datasets we evaluate on, including some of the data in our test splits, so we ignore it when averaging over models.

over different answers can mitigate this problem. We use a variant of the calibration method presented in Zhao et al. (2021) by balancing predictions to be 50/50 across the two output labels. Specifically, if $l_+$ and $l_-$ are the logits for the positive and negative label respectively, then instead of classifying an example as positive if $l_+ > l_-$, we classify it as positive if $l_+ > l_- + \gamma$, where we select the threshold $\gamma \in \mathbb{R}$ so that the predictions are balanced. We find this increases accuracy by about 5% on average. Unless otherwise specified, we always report zero-shot accuracy after calibration.

Encoder-only models (e.g. RoBERTa and DeBERTa) cannot be easily used to do zero-shot classification out of the box, so to evaluate them we follow the method of Yin et al. (2020): we finetune both models on an NLI dataset (MNLI, which we do not evaluate on) and treat the difference between the entailment and contradiction probabilities as the effective logit. This provides a strong zero-shot baseline for encoder-only models that works even for non-NLI tasks (Yin et al., 2020). This finetuning isn't necessary for CCS to work on encoder-only models (see Appendix C), but we test CCS using the same MNLI-finetuned models for ease of comparison.

**Hidden states.** We extract the hidden states corresponding to the last token in the last layer of each model for simplicity, unless otherwise specified. For encoder-decoder models, we evaluate CCS on the last layer hidden states of both the encoder and decoder, and use whichever one generally achieves a lower unsupervised loss; for T0 this is the decoder hidden states, while for T5 and UnifiedQA this is the encoder hidden states. See Appendix G.2 for further implementation details, such as tokenization.

**Prompts.** To reduce prompt sensitivity, we use between 8 and 13 prompts for each dataset (9 on average), derived or slightly modified from Sanh et al. (2021). Unless otherwise specified, we average across all prompts when showing results. To construct contrast pairs, we let $x_i^+$ be the zero-shot prompt using $q_i$ and the first label (e.g. "Positive" for sentiment classification datasets) and let $x_i^-$ be the prompt using $q_i$ and the second label (e.g. "Negative"). We describe all prompts in Appendix I.

## 3.2 EVALUATING CCS

### 3.2.1 CCS OUTPERFORMS ZERO-SHOT

We evaluate CCS on all 6 models and compute the average accuracy across all datasets and prompts. T0 was trained on 9 out of 10 of the datasets we evaluate on, including some of the data in our test splits, so we ignore it when averaging over models to avoid unfair comparisons. We display the results in Table 1. To assess prompt sensitivity, for each model and dataset we compute the standard deviation (s.d.) of accuracy across different prompts, then average the resulting standard deviations across all datasets, which we show in parentheses in Table 1. For comparison, we also include results when training CCS on all datasets simultaneously, which we refer to as CCS (All Data).

CCS attains an accuracy of 71.2% on average, compared to 67.2% for calibrated zero-shot. It outperforms zero-shot accuracy for every model, except for RoBERTa (where it does 2% worse) and T0 (for which zero-shot accuracy is inflated). Training on all datasets improves accuracy by only an insignificant amount on average (0.3%), but with large gains for T0 in particular (77.6% → 84.8%).

These results show that CCS can *exceed* the performance of strong baseline methods that access the model outputs, even though this wasn't our main goal. This indicates that it is indeed possible to classify examples with high accuracy using only unlabeled model representations.

### 3.2.2 CCS Is Robust To Misleading Prompts

Recall our goal: to discover latent knowledge in a language model even when the model outputs false text. In particular, language models are typically trained to imitate text whether or not it is correct, so if a model sees false text it should intuitively be more likely to predict that subsequent text will also be false. Based on this idea, we provide an initial proof of concept that CCS can make progress toward our goal by constructing prompts that aim to mislead the outputs of language models. Specifically, we add a prefix to the beginning of our zero-shot prompts that consists of questions answered incorrectly (Figure 5). The hope is that such a prefix will decrease zero-shot accuracy because the model will imitate its context and answer subsequent questions incorrectly even if it internally "knows" better.[2] We found that while most models are robust to this type of prefix (see Appendix B), it significantly drops calibrated zero-shot performance in UnifiedQA, decreasing accuracy from $80.4\%$ to $70.9\%$.

We evaluate CCS on these examples and show the results in Figure 4 of the Appendix. We find that despite the $9.5\%$ drop in zero-shot accuracy, CCS maintains high accuracy ($82.1\% \rightarrow 83.8\%$). This provides evidence that our method can still work well even when model outputs are unreliable.

## 3.3 Analyzing CCS

We have shown that CCS attains strong classification performance in standard settings and when we deliberately mislead models. Moreover, we have described our motivation as discovering latent representations of truth in language models, but in practice CCS just finds a direction in representation space that attains high accuracy. This raises the question: in what sense is CCS actually finding "truth" features? We now provide a preliminary investigation of this question.

### 3.3.1 CCS Finds A Task-Agnostic Representation of Truth

From the results described so far, it may be possible that the classifier we find is capturing dataset-specific properties, such as which of two labels (e.g. "Yes" vs. "No") is more likely to be correct. We rule this out by showing that it generalizes across completely different tasks, including ones with different label spaces (such as from "Yes" and "No" for a generic task to "Positive" and "Negative" for sentiment classification). In particular, we train and test CCS on every pair of datasets, and show the resulting transfer for several models in Figure 2. (See Appendix F for results with other models.)

We find that CCS indeed transfer wells: in the majority of datasets, the transfer accuracy of CCS is competitive with training and testing on the same dataset (the last row in Figure 2, "No transfer"). Transfer performance can even outperform the no-transfer setting, especially when we train on simpler tasks, such as sentiment classification. For example, training CCS on the Amazon sentiment dataset achieves an average transfer accuracy of $71.8\%$, which is $0.6\%$ higher than CCS without transfer. We speculate that this performs so well because the difference in representations between correct and incorrect answers is especially pronounced for easier tasks like Amazon.

Additionally, transfer accuracy tends to be similar for many datasets. For example, training on IMDB (row 1 of Figure 2) has similar accuracy as training on DBPedia (row 4). This provides evidence that CCS can find a functionally similar direction across many different types of training datasets. Overall, these results suggest that (1) models may have a task-agnostic representation related to what is true, and that (2) CCS may approximately find this representation even without diverse data.

### 3.3.2 CCS Does Not Just Recover Model Outputs

One possibility is that CCS can only recover knowledge already contained in a model's outputs. We have shown that CCS can outperform zero-shot accuracy (Section 3.2.1), especially when model outputs are misled (Section 3.2.2), which already provides evidence against this possibility. We now provide additional evidence against this concern.

First, if CCS were just recovering knowledge in the model outputs, using the last layer of a network (right before the outputs) should presumably outperform intermediate layers (which are more causally distant from the outputs). However, for T5 and UnifiedQA, we find that using hidden states in the

---

[2] In practice we found that model behavior was qualitatively similar between using this prefix with correct and incorrect answers, in agreement with the findings of Min et al. (2022b) that using incorrect labels in demonstrations often does not significantly degrade the performance of a few-shot prompt. Consequently, the prefix may instead actually be reducing accuracy because it is out-of-distribution. See also Kim et al. (2022) for more experiments on the subtleties of correct vs incorrect labels for few-shot prompts.

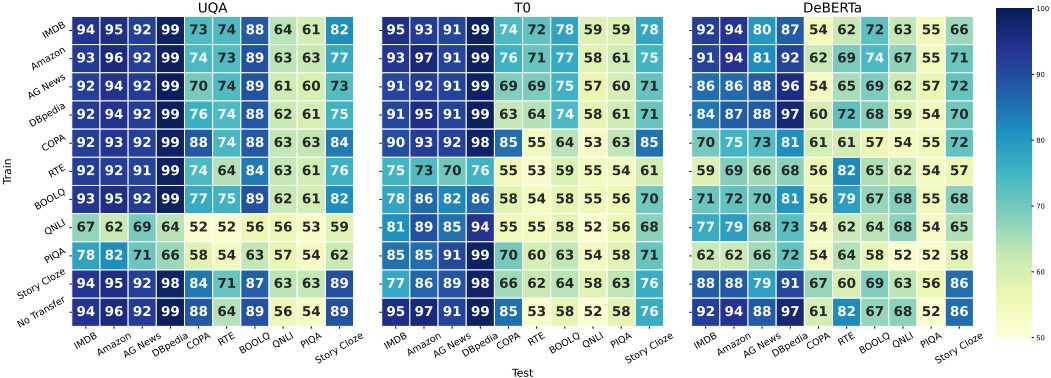

Figure 2: Transfer Performance using CCS on UnifiedQA, T0 and DeBERTa. The y-axis corresponds to the training dataset, and the x-axis corresponds to the test dataset. The final row ("No Transfer") corresponds to training and testing on the same dataset, which is the same as the diagonal. On most datasets, CCS transfers well to other datasets (relative to no transfer), including to different tasks with completely different labels. In some cases transfer even outperforms the no-transfer setting. See Appendix E for results with other models.

middle of the network outperform hidden states at the end of the network when using CCS (see Figure 10). This is especially true for UnifiedQA on misleading prefixes; we find that using the encoder hidden states is robust to misleading prefixes (Section 3.2.2), but that accuracy using the decoder hidden states drops from $81.0\%$ to $73.5\%$, a similar amount to zero-shot accuracy. This suggests that compared to the later layers of a model, intermediate layers are more robust and less correlated with the model outputs, and that CCS can take advantage of this.

Finally, if CCS were just recovering knowledge in the model outputs, we would only expect it to work in cases where model outputs are informative. However, we show in Appendix C that CCS still works with masked language models when their outputs are uninformative: when we don't [MASK] any input tokens, and when we prompt models so that the labels used to construct contrast pairs appear in the *middle* of a prompt rather than at the end. These results show that CCS can sometimes recover latent knowledge in a model that is distinct from—and more useful than—what the model outputs.

### 3.3.3 TRUTH IS A SALIENT FEATURE

From the results we have presented so far, it is possible that the direction learned by CCS is difficult to find and requires using a large amount of unsupervised data. We provide evidence against this possibility by showing that finding such a direction can both (1) often be done with only a small amount of data, and can also (2) often be done by essentially taking the top principal component of a slightly modified representation space.

**CCS doesn't require much data.** We now evaluate how well CCS performs with different amounts of data. Whereas before we trained CCS using the full training set and all prompts, here we use limited data and a single prompt. Specifically, we train using only $k$ unlabeled contrast pairs, using the single prompt for each model and dataset that achieves the highest zero-shot accuracy. We still test on all prompts for each dataset. We resample $k$ points 32 times for each of $k = 1, 2, 4, \cdots$, and take the average accuracy across those 32 samples. Finally, we plot the average such accuracy across all datasets and prompts for several models in Figure 3.

We find that while CCS benefits from more data, it can often do well with very limited data. In fact, it can sometimes even do well with only a single contrast pair, though we find high variance across individual datasets; see Appendix D for more details. This suggests that the strong performance of CCS does not primarily come from using a large amount of unsupervised data, and indicates that the direction learned by CCS may be relatively easy to find.

**Contrastive Representation Clustering.** We now show that directions correlated with the truth may be "salient" in a different way: by showing that we can also find such directions using either (1) PCA or (2) clustering. Specifically, suppose we construct contrast pairs $(x_i^+, x_i^-)$ as before. Intuitively, these two examples are qualitatively almost identical except that one is true and the other

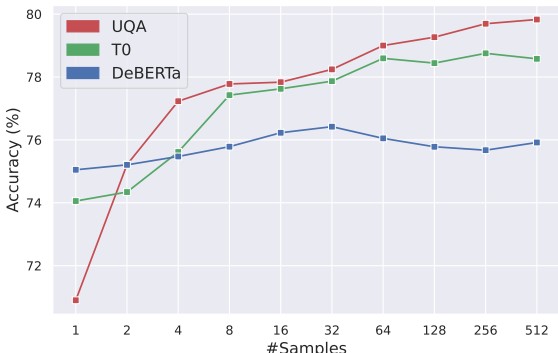

Figure 3: Accuracy when we train CCS on $k$ samples for different values of $k$ (each time averaged across 32 trials). We use the single prompt with the highest zero-shot accuracy for each dataset and model. While CCS benefits from more examples, it can often work well with limited data.

is false, so the main difference between the representations $\tilde{\phi}(x_i^+)$ and $\tilde{\phi}(x_i^-)$ should relate to truth. Consequently, we can take the differences in (normalized) hidden states, $\{\tilde{\phi}(x_i^+) - \tilde{\phi}(x_i^-)\}_{i=1}^n$, and cluster them. We call this method Contrastive Representation Clustering (CRC). Clustering can be achieved by taking the top principal component (TPC) and thresholding at 0, or by doing a "bimodal salience search" (BSS) to find a direction that looks bimodal; see Appendix G.4 for further details.

We compare CCS and these two variants of Contrastive Representation Clustering in Table 2, using the same setting as in Table 1. While CCS performs best, all methods attain high accuracy and are competitive with zero-shot performance. This indicates both that (1) representations of truth often lie in a high-variance direction in the contrastive representation space $\{\tilde{\phi}(x_i^+) - \tilde{\phi}(x_i^-)\}_{i=1}^n$, and also that (2) true and false examples are often well-clustered in this same contrastive space. This strengthens the idea that representations of truth may be salient features inside models that are relatively easy to find. This may help explain why CCS can perform well without using any supervision, and how it can do so even with only a limited amount of unlabeled data.

## 4    RELATED WORK

**Zero-Shot Prompting.** Since the release of GPT-3 (Brown et al., 2020), one of the main paradigms for eliciting what models know has been zero-shot prompting (Liu et al., 2022; Beltagy et al., 2022). Zero-shot exploits how language models are trained to predict diverse data from the internet, which incidentally includes tasks such as question-answering. If prompted appropriately, this can be used to solve various useful tasks with reasonable performance (Brown et al., 2020). However, these models are trained to imitate human-generated data, which bounds the quality of their outputs.

Many methods improve upon vanilla zero-shot prompting (Liu et al., 2022; Zhao et al., 2021; Lu et al., 2022; Wei et al., 2022b; Min et al., 2022a). While our goal is not to improve zero-shot performance, some of the ideas underlying these methods are similar to CCS. Particularly relevant are methods that also leverage unsupervised consistency properties, such as Jung et al. (2022); Zhou et al. (2022). However, these methods still bootstrap from language model outputs trained via imitation learning, which limits their applicability to our main goals.

| Method | RoBERTa | DeBERTa | GPT-J | T5 | UQA | T0* | Mean* |
|---|---|---|---|---|---|---|---|
| Calibrated 0-shot | 64.3(6.2) | 76.3(6.0) | 56.0(5.2) | 58.8(6.1) | 80.4(7.1) | 90.5(2.7) | 67.2(6.1) |
| CCS | 62.1(4.1) | **78.5(3.8)** | **61.7(2.5)** | **71.5(3.0)** | **82.1(2.7)** | 77.6(3.3) | **71.2(3.2)** |
| CRC (TPC) | **64.6(5.8)** | 76.5(5.7) | 59.9(3.7) | 66.7(5.0) | 78.3(3.3) | 58.9(13.0) | 69.2(4.7) |
| CRC (BSS) | 62.6(6.8) | 76.8(5.3) | 60.7(3.4) | 69.7(3.3) | 79.3(2.5) | 76.7(9.6) | 69.8(4.3) |

Table 2: We compare CCS to two variants of Contrastive Representation Clustering: TPC, which clusters by projecting onto the top principal component, and BSS, which clusters by finding a direction that looks bimodal. We show accuracy and standard deviation of each model averaged across all prompts and datasets, in the same setting as Table 1. We find that CCS generally performs the best, but that all methods are competitive with zero-shot.

To illustrate this, imagine we train reinforcement learning agents to play a game such as Diplomacy (FAIR et al., 2022), in which players have incentives to lie to each other. Then those agents may learn to lie in a way that is difficult to detect, but they may still internally represent whether they are lying. Because their outputs would be deliberately misleading, standard zero-shot methods may be very unreliable. In contrast, techniques like CCS may still be able to detect whether those models are lying by finding representations of truth in their activations that contradict their outputs.

**Truthfulness.** There has been increasing interest in making language models truthful (Evans et al., 2021; Lin et al., 2022). One aspect of truthfulness that has received substantial attention is factuality (Thorne et al., 2018; Maynez et al., 2020). For instance, many techniques aim to improve the factuality of models by augmenting them with retrieval methods (Nakano et al., 2021; Menick et al., 2022), which allows them to cite their sources. In contrast, we focus on truthfulness more generally, which also includes procedural knowledge such as reasoning or natural language inference tasks.

An approach to making language models truthful in this more general setting is to finetune them using either human demonstrations (Khashabi et al., 2020; Sanh et al., 2021; Zhong et al., 2021; Wei et al., 2022a) or reinforcement learning from human feedback (Christiano et al., 2017; Stiennon et al., 2020; Askell et al., 2021; Bai et al., 2022; Ouyang et al., 2022). These techniques have been widely successful at improving performance, but unlike our method they rely on being able to provide ground truth labels, which can be intractable in many settings.

Some work has aimed to go beyond the direct supervision humans can provide by augmenting supervision with AI systems (Christiano et al., 2018; Irving et al., 2018; Leike et al., 2018; Perez et al., 2022). This may expand the range of applications we can supervise, but many of these proposals remain theoretical, and it is unclear just how far these techniques can generalize. Christiano et al. (2022) reframes this issue by posing the problem of Eliciting Latent Knowledge (ELK). Like our problem statement, ELK is about eliciting knowledge from models even in cases where humans cannot evaluate that knowledge. However, ELK frames this as a worst-case theoretical problem, while we frame this as an empirical problem that we can make progress on using current models.

## 5 DISCUSSION

### 5.1 LIMITATIONS AND FUTURE WORK

Our work has a number of limitations. First, CCS relies on the existence of a direction in activation space that separates true and false inputs well, in the sense that a supervised probe on the activations would be able to attain high accuracy (if it hypothetically had access to the ground truth labels). This requires that a model is both *capable* of evaluating the truth of a given input, and also that the model *actively evaluates the truth* of that input. It is not clear when these conditions hold precisely.

Second, we did not evaluate our method on setups involving active "lying" or "deception" (Kenton et al., 2021; Evans et al., 2021) by models, as we aren't aware of existing evaluation setups for this setting. If future work develops such a setup, a good stress test would be to apply CCS to do "lie detection" in that setting. This may require modifications or extensions to the method, such as more explicitly ensuring that it recovers the truth of an input rather than what the model says.

There are also various straightforward improvements to our method that one could explore. This includes adding additional consistency constraints, improving its reliability, calibrating its probabilities, generalizing it beyond the yes-no question-answering setting, generalizing it to cases where answers aren't clear-cut, and closing the remaining gap between CCS and the logistic regression ceiling.

### 5.2 CONCLUSION

As language models become more capable, they will be increasingly used as components in larger AI systems trained with reinforcement learning. As this occurs, falsehoods arising from misaligned training objectives may become more common, severe, and difficult to detect. In principle, models may even develop instrumental incentives to lie: for example, if a human evaluator would disapprove of bad model behavior, models may learn to lie about their behavior to achieve higher reward. If so, those models would be optimizing against human evaluators, making it precarious to rely on those evaluators for assessing the truth of what models say. Because unsupervised methods for eliciting answers circumvent this issue, they may still work even in this scenario. We show that it is possible to make progress on such methods today; the empirical success of our approach suggests that unsupervised methods are both a tractable and underexplored research direction.

ACKNOWLEDGEMENTS

We are very grateful to Jared Kaplan for helpful experiment suggestions and resources early on in the project. We thank Beth Barnes and Paul Christiano for valuable discussions regarding the longer-term impacts of this work. We are also grateful to Jessy Lin, Alex Pan, Ruiqi Zhong, Yaodong Yu, the anonymous reviewers, and several others for useful feedback on earlier versions of this paper. CB is supported by an Open Philanthropy AI Fellowship.

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

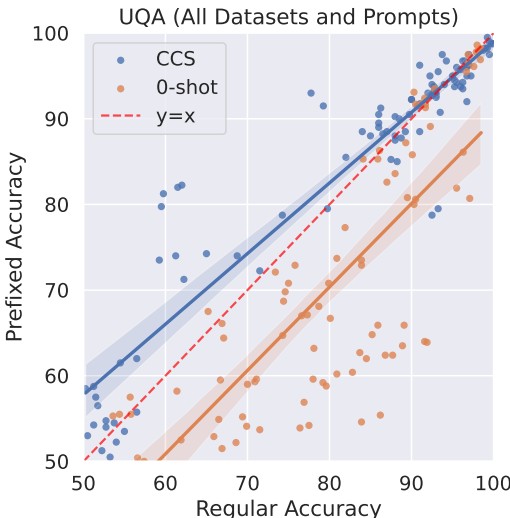

Figure 4: Zero-shot accuracy drop caused by the misleading prefix shown in Figure 5 for UnifiedQA. Each point corresponds to one dataset and one prompt. On average, zero-shot accuracy drops $9.5\%$, while CCS performs $1.7\%$ better. This suggests that the latent knowledge inside a model that CCS recovers can be robust even when model outputs become less reliable.

## A    IDENTIFYING CLUSTERS

CCS predicts yes-no answers to a set of unlabeled questions $q_1, \ldots, q_n$. However, it does not identify which prediction label corresponds to true and which corresponds to false. We now describe how one can do so in principle, as long as a model is also able to take conjunctions.

Suppose we run our method and find, for example, that $x$ and $x'$ end up in different clusters. Then we know that these two statements have opposite truth values. As a result, if we take the conjunction of these two statements, $x \wedge x'$ then it should be false, and if we take the disjunction, $x \vee x'$, then it should be true. This allows us to identify which cluster is true and which is false in a completely unsupervised way.

## B    MISLEADING PREFIX DETAILS

### B.1    MISLEADING PREFIX

In Section 3.2, we mentioned that model outputs can be biased by using the misleading few-shot prefix shown in Figure 5. We show the effect of the misleading prefix for all models in Table 3, and we illustrate the drop in accuracy for each dataset and prompt in Figure 4.

### B.1.1    EVALUATING THE EFFECT OF THE MISLEADING PREFIX

The hope is that the misleading prefix causes models to imitate the sorts of incorrect answers in the prefix. However, we found that results are qualitatively similar with correct answers, making the actual interpretation of the effect of this prefix ambiguous. Nevertheless, to better understand what this prefix is doing, we visualize the top 100 tokens for both this misleading prefix and no prefix on average across all examples for UnifiedQA on the IMDB sentiment dataset, and show the results in Figure 6. Without this prefix, the top two tokens are the actual labels ('positive' and 'negative'). In contrast, with the prefix, these tokens have lower probabilities relative to other tokens, with the highest probability tokens instead being irrelevant to the true answer. For example, since the prefix includes several answers with numbers, many of the top tokens are also numbers. This provides some evidence that the prefix is actually causing the model to output false text by imitating its context in a meaningful sense.

What is the human life expectancy in the United States?
10 years.

Who was president of the United States in 1955?
Abraham Lincoln.

Which party did he belong to?
The Social Democratic Party of Germany.

What is the square root of banana?
42

How does a telescope work?
Eye beams are emitted by the eye and reflect back into the eye.

Where were the 1992 Olympics held?
The 1992 Olympics were held on the moon.

How many squigs are in a bonk?
3

Figure 5: The added prefix used to mislead models in a way that is egregiously incorrect or confusing.

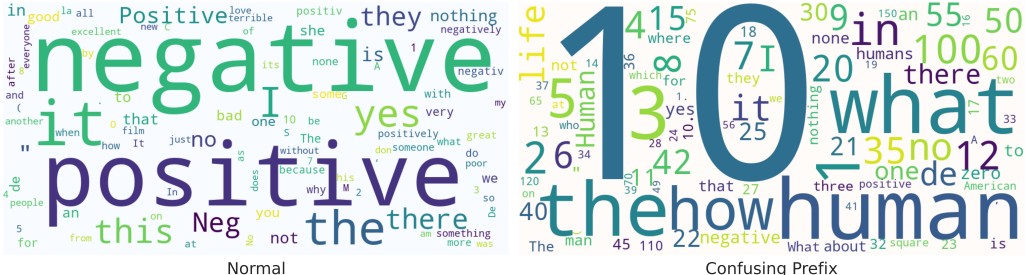

Normal                    Confusing Prefix

Figure 6: The top 100 tokens with highest probability averaged across examples in IMDB for UQA. Without the prefix (left), the actual labels ("positive" and "negative") and their synonyms (e.g. "Positive" and "Neg") have high probability. With the prefix, the model's output becomes mostly irrelevant to sentiment; the actual labels are in the top 100, but no other synonyms are.

| Method | Prefix | RoBERTa | DeBERTa | GPT-J | T5 | UQA | T0* | Mean* |
|--------|--------|---------|---------|-------|-----|-----|-----|-------|
| Calibrated | Regular | 64.3(6.2) | 76.3(6.0) | 56.0(5.2) | 58.8(6.1) | 80.4(7.1) | 90.5(2.7) | 67.2(6.1) |
| 0-shot | Prefix | 65.6(5.3) | 75.5(6.1) | 59.2(4.6) | 56.0(4.0) | 70.9(7.8) | 88.2(4.2) | 65.4(5.6) |
| CCS | Regular | 62.1(4.1) | 78.5(3.8) | 61.7(2.5) | 71.5(3.0) | 82.1(2.7) | 77.6(3.3) | 71.2(3.2) |
| | Prefix | 62.2(3.6) | 75.4(5.2) | 61.2(1.8) | 73.2(2.6) | 83.8(2.4) | 75.0(2.7) | 71.2(3.1) |
| TPC | Regular | 64.6(5.8) | 76.5(5.7) | 59.9(3.7) | 66.7(5.0) | 78.3(3.3) | 58.9(13.0) | 69.2(4.7) |
| | Prefix | 65.0(5.6) | 77.0(6.0) | 60.1(2.9) | 68.1(5.0) | 76.0(3.6) | 56.5(13.6) | 69.2(4.6) |
| BSS | Regular | 62.6(6.8) | 76.8(5.3) | 60.7(3.4) | 69.7(3.3) | 79.3(2.5) | 76.7(9.6) | 69.8(4.3) |
| | Prefix | 59.5(5.4) | 75.5(6.6) | 59.8(2.8) | 65.2(2.8) | 83.1(2.3) | 72.2(9.0) | 68.6(4.0) |
| LR | Regular | 79.8(2.5) | 86.1(2.2) | 78.0(2.3) | 84.6(3.1) | 89.8(1.9) | 90.7(2.1) | 83.7(2.4) |
| | Prefix | 79.4(2.7) | 86.3(2.6) | 79.0(2.9) | 86.3(2.8) | 90.1(2.2) | 90.5(2.3) | 84.2(2.6) |

Table 3: Accuracy of each method and model averaged across all prompts and datasets, with the same setting as Table 1. "Regular" means no prefix is added, while "Prefix" corresponds to text in Figure 5. Notice that while the 0-shot accuracy decreases due to the prefix, all of our methods are more resistant to the influence.

# C  MASKED LANGUAGE MODELING RESULTS

We now provide an initial demonstration that CCS can work well even when a model's outputs are not very useful.

First, we evaluate our method on a model trained exclusively with the masked language modeling (MLM) objective: DeBERTa-v2 (He et al., 2021) *without* NLI finetuning (unlike in Section 3, where we finetuned DeBERTa on an NLI task to use it in the zero-shot setting). The outputs of DeBERTa are unlikely to be meaningful if we provide a raw input without using any [MASK] tokens[3]. If CCS only works when the model outputs are useful, then it should perform poorly in this setting.

We also consider one other way that model outputs can be uninformative. Given a sequence of tokens as input, DeBERTa has a different set of output logits for each input token. For a given contrast example (a question and a candidate answer), the most informative output should intuitively be the logits that predict the answer. For instance, if the contrast example $x^+$ is "Is 2+2=4? Yes", then the most informative output should be the logits corresponding to candidate answer "Yes". Consequently, if we instead format each example so that the label (e.g. "Yes" or "No") appears in the *middle* of the prompt, then the output corresponding to the *final* token should not be very informative. If CCS only works well when the model outputs are useful, then it should also perform poorly in this setting (as long as CCS uses the last-token hidden states as usual).

To show that CCS can work even when model outputs aren't informative, based on the above discussion we now do an initial test of CCS when we simultaneously (1) use DeBERTa-v2 (MLM-pretrained only), and (2) format inputs so that the label is in the middle of the prompt rather than at the end. In particular, we apply CCS on the Amazon dataset, where we randomly sample 1000 new (unlabeled) points and use a 60/40 train-test split as before. As usual we continue to use the last-token hidden states for CCS. As usual, we use the default Huggingface tokenizer and we do not [MASK] any input tokens. We use the following custom prompt to test (2):

```
The following movie review expresses a [label] sentiment:\n[text]
```

Even though it is unclear why the model outputs should be useful with this setup, we find that CCS can indeed still perform well, attaining an accuracy of 93.7%. For comparison, this is nearly identical to the approximately 94% accuracy of CCS when evaluated on the Amazon dataset using NLI-finetuned DeBERTa and the original prompts (see Figure 9).

We argued above that the model outputs should not be very useful in this setting. We now verify that this is the case. To do so, we evaluate a modified version of zero-shot prompting adapted to work for masked language models when we use no [MASK] tokens in the input. Specifically, on an input $x^+$ or $x^-$, we (1) take the logit vector corresponding to the last input token, (2) select the corresponding logit for "positive" and the corresponding logit for "negative" (the two candidate labels), (3) take the difference between these, resulting in $l^+_{pos} - l^+_{neg}$ for $x^+$ and $l^-_{pos} - l^-_{neg}$ for $x^-$, respectively, then (4) take the difference between these, $(l^+_{pos} - l^+_{neg}) - (l^-_{pos} - l^-_{neg})$, to form the effective logit. Intuitively, this final expression should be large if the example is positive, and small if the example is negative. As before, we calibrate this zero-shot method so that its predictions are balanced.

We evaluate this method and find that it recovers non-trivial knowledge in the model outputs, but that it still performs substantially worse than CCS: calibrated zero-shot accuracy in this setting is 71.6%, compared to almost 94% for CCS. This suggests that model outputs in this setting indeed aren't very useful, especially relative to CCS.

Overall, our results in this section show that CCS can still work in at least some settings where model outputs don't seem to be very useful. This provides additional evidence that out approach does not simply recover knowledge represented in the model outputs.

---

[3]The MLM objective first selects 15% of tokens to mask. Of the masked tokens, 80% are replaced with [MASK], 10% are randomly replaced with a different token, and 10% are left unchanged (He et al., 2021). As a result, for non-[MASK] tokens, the most likely output token should usually be the original input token itself, which is uninformative.

# D  COMPLETE SAMPLE COMPLEXITY RESULTS

In this section we show additional results on how the number of examples affects CCS. We show the performance (averaged across all datasets) for *all* models in Figure 7. We next show a more fine-grained results. Specifically, we select #Samples = 1, 8, 64 and show dataset-level results in Figure 8 for each model.

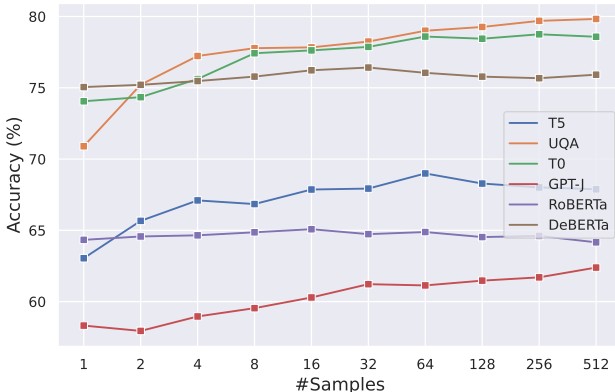

Figure 7: This figure shares the same setting with Figure 3, but it contains all six models we consider.

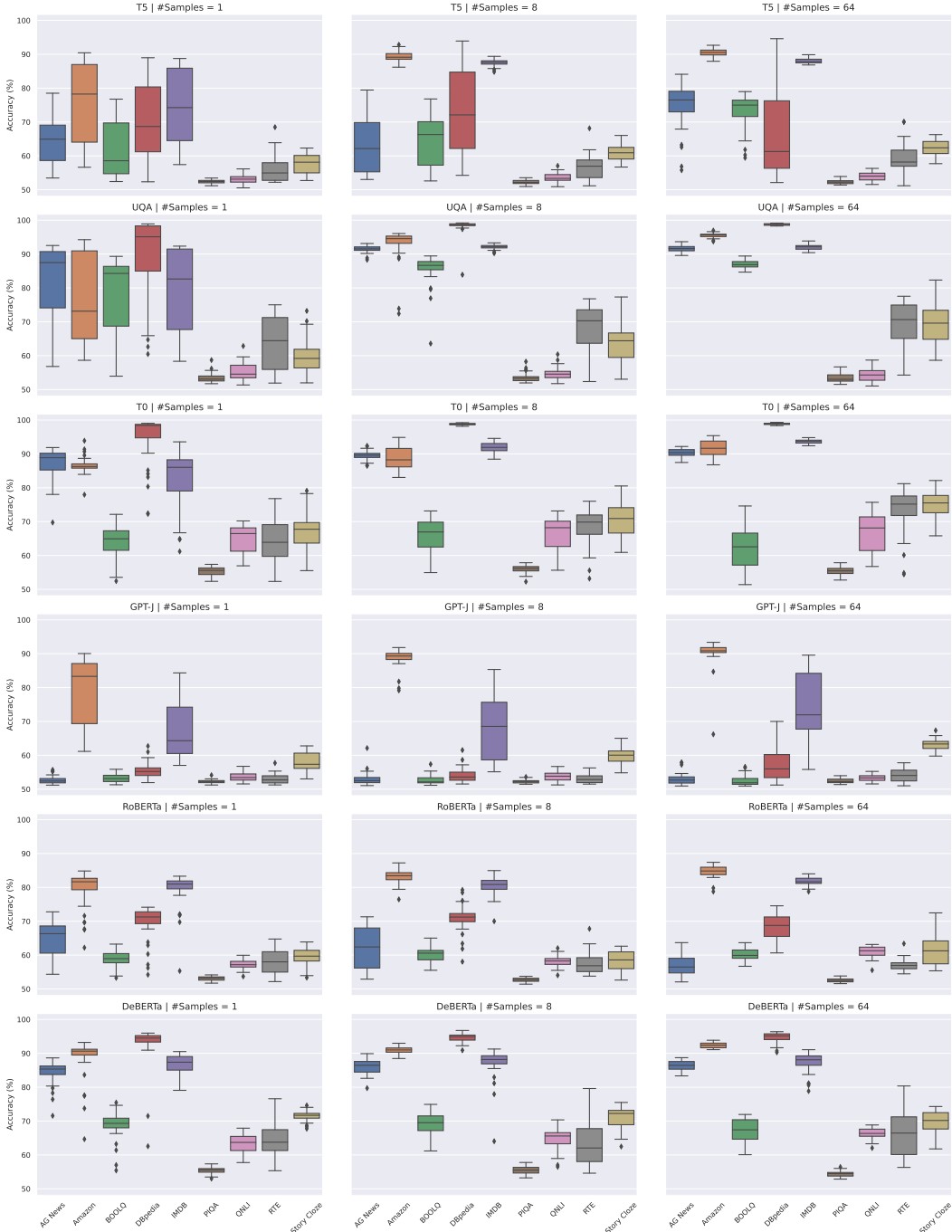

Figure 8: CCS performance when we only use $1, 8, 64$ examples, for all models and datasets. X-axis represents the dataset and y-axis is the accuracy in percentage. For each dataset, we select the prompt with highest 0-shot performance, and then randomly select $1, 8, 64$ data points from this prompt. We then perform CCS and test on all prompts of this dataset, where each value in the barplot corresponds to one prompt.

# E  COMPLETE TRANSFER RESULTS

Complete transfer results for CCS, TPC and BSS in all models are shown in Figure 9.

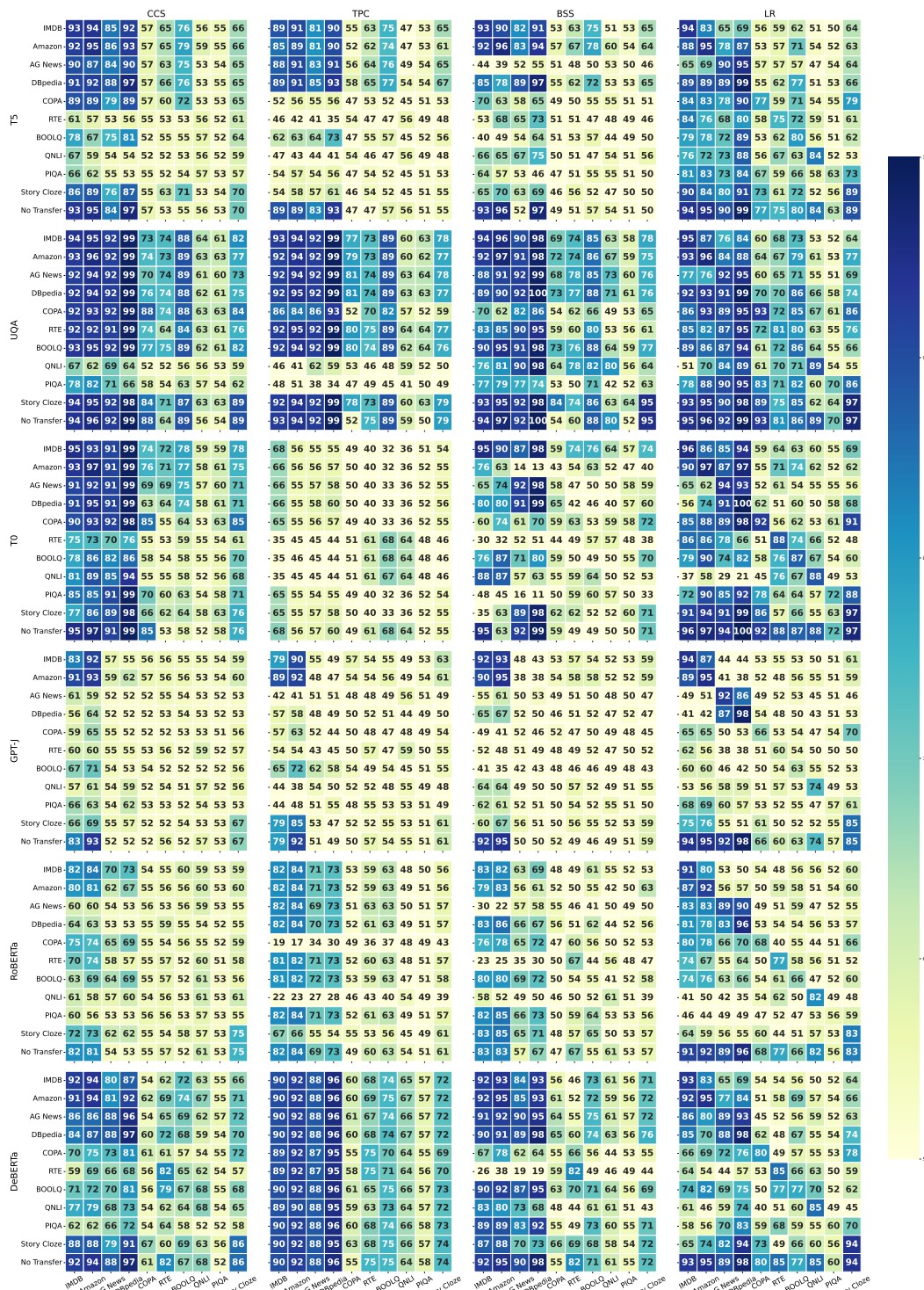

Figure 9: Transfer Accuracy for all models using CCS, BSS, TPC and LR.

# F COMPLETE INTERMEDIATE REPRESENTATIONS RESULTS

In this section we the performance of CCS and LR across all hidden layers in all six models. Specifically, for each model, we generate the hidden states for each dataset every 2 layers (for both encoders and decoders). Then, for each set of hidden states, we perform CCS and LR on each dataset separately, and average the performance across all datasets. We show CCS results in Figure 10 and LR results in Figure 11.

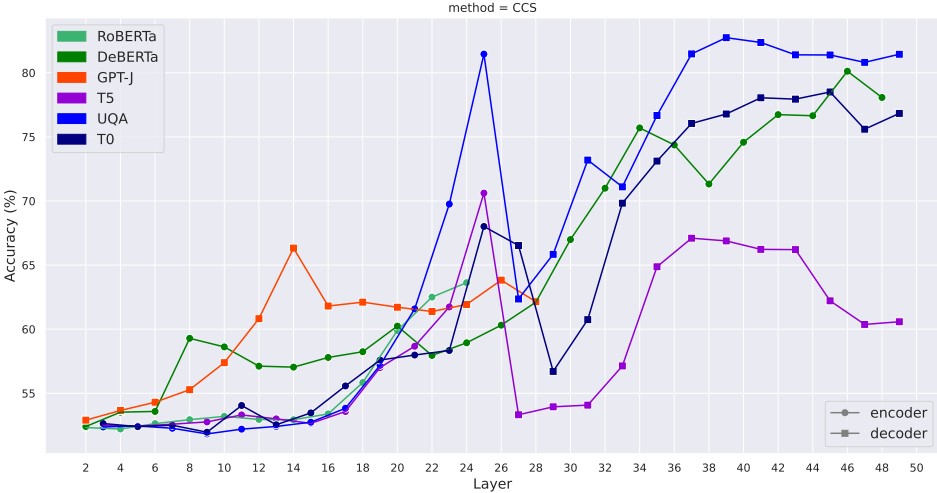

Figure 10: CCS performance when using the hidden states across different layers, using all six models we consider in the paper.

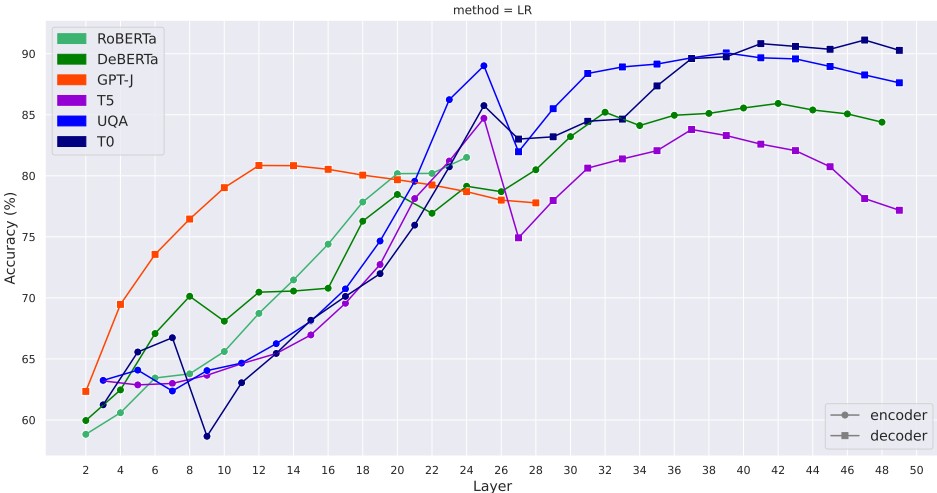

Figure 11: Linear regression performance when using the hidden states across different layers, using all six models we consider in the paper. This is the ceiling of all possible methods, and is supervised.

# G  CCS AND CRC IMPLEMENTATION DETAILS

In this section, we provide further implementation details for CCS and CRC.

## G.1  NORMALIZATION

As described in Section 2.2, we normalize the features $\tilde{\phi}(x_i^+) = \phi(x_i^+) - \mu^+$, where $\mu^+$ is the mean of $\{\phi(x_i^+)\}_{i=1}^n$ (and similarly for $\tilde{\phi}(x_i^-)$). In practice, we also normalize the scale of the features by also dividing by the average norm of $\{\phi(x_i^+)\}_{i=1}^n$ times $\sqrt{d}$ (and similarly for $\tilde{\phi}(x_i^-)$). However, this is less essential than mean normalization and there are likely many reasonable choices for scale normalization.

## G.2  CONTRAST PAIRS

We use the Huggingface library (Wolf et al., 2019) for all of our experiments. We use the standard tokenizer for each model, and always take the hidden state corresponding to the last token in a given layer indexed by idx. We show Huggingface-style pseudocode for extracting hidden states from in Algorithm 1.

For encoder-only and decoder-only models, we provide the full input, $x^+$ or $x^-$, which includes both the question *and* the proposed answer, to the model; see Appendix I.1 for more formatting and tokenization details. For encoder-decoder models, our input format depends on whether we are taking the encoder hidden states or the decoder hidden states. When we take the decoder hidden states, we input the question to the encoder, and input the candidate answer to the decoder. In contrast, when we take the encoder hidden states of an encoder-decoder model, we provide the full input (including the proposed answer) to the encoder, and ignore the decoder (simply passing the empty string to the decoder and ignoring its results). This is necessary to ensure that the inputs to the encoder are not identical across contrast pairs.

---

**Algorithm 1** Pseudocode for Getting Contrast Features

---

**Input:** Contrast Pairs Set $S$, model name mdl, layer index idx
   $\mathbf{m}$ = transformers.AutoModel(mdl)
   $\mathbf{t}$ = transformers.AutoTokenizer(mdl)
   $\mathcal{C}^+, C^- = [], []$
   **for** $(x^+, x^-)$ **in** $S$ **do**
      $\text{token}^+, \text{token}^- = \mathbf{t}.\text{encode}(x^+), \mathbf{t}.\text{encode}(x^-)$
      $\phi^+ = m(\text{token}^+, \text{output\_hiddenstates} = \text{True})[\text{``hidden\_states''}][\text{idx}][-1]$
      $\phi^- = m(\text{token}^-, \text{output\_hiddenstates} = \text{True})[\text{``hidden\_states''}][\text{idx}][-1]$
      $\mathcal{C}^+.\text{append}(\phi^+), \mathcal{C}^-.\text{append}(\phi^-)$
   **end for**
**Output:** $[\mathcal{C}^+, \mathcal{C}^-]$

---

## G.3  CCS

Given contrast features from Algorithm 1, CCS works by learning parameters $\theta$ and $b$ by minimizing $L_{CCS}(\theta, b)$ as defined in Section 2.2.

In practice, we implement the bias $b$ by appending an additional dimension of 1s features to the input, increasing it and $\theta$ to have dimension $d + 1$. We randomly initialize $\theta$ to have unit norm. We then optimize the CCS loss $T = 10$ times and select the run with lowest unsupervised loss. In practice, we train each time for $E = 1000$ epochs with a learning rate $\eta = 0.01$ (which we found was good for consistently achieving low unsupervised loss) in each run. Because we are only learning a linear probe on top of the features, training can be fast.

## G.4 CRC: TOP PRINCIPAL COMPONENT

Intuitively, a direction that clusters examples well should have high variance. Motivated by this, a simple method is to cluster according to the Top Principal Component (TPC). We do this by first constructing contrast features from the normalized contrast pair activations, $\mathcal{C} = \{\tilde{\phi}(x_i^+) - \tilde{\phi}(x_i^-)\}_{i=1}^n$, then projecting these examples on their top principal component using PCA. We then treat examples that are less than 0 as one cluster and examples that are greater than 0 as the other cluster. We find that despite its simplicity this method can often also find truth-like features in model activations. This method is similar in spirit to Bolukbasi et al. (2016), except that our method doesn't require any labels; we leverage the fact that truth is consistent to construct contrast pairs in a purely unsupervised way.

## G.5 CRC: BIMODAL SALIENCE SEARCH

Bimodal Salience Search (BSS) is another variation of CRC. One drawback of TPC is that variance is not an intrinsic quantity to a network's behavior, since different directions in representation space can be scaled without changing the behavior of the network as long as subsequent layers are rescaled accordingly. This motivates using a method that clusters examples regardless of the scale of different directions.

If examples are well-clustered, then the intra-cluster variance should be low while the inter-cluster variance (or total variance) should be high. Specifically, if we center a set of datapoints so that 0 delineates the cluster boundary, then the points on either side of 0 should have low variance compared to the points overall. This suggests minimizing the following loss:

$$L(\theta) = \frac{\text{var}\{\theta^T c_i | \theta^T c_i < 0\} + \text{var}\{\theta^T c_i | \theta^T c_i \geq 0\}}{\text{var}\{\theta^T c_i\}} \tag{1}$$

where $c_i := \tilde{\phi}(x_i^+) - \tilde{\phi}(x_i^-)$ and where we use var$\{z_i\}$ as shorthand to denote the variance of a set $\{z_i\}_{i=1}^n \subset \mathbb{R}$. Because this objective is written in terms of a ratio, it is invariant to the overall scale of the direction, fixing that drawback of TPC. This loss is similar to that of Linear Discriminant Analysis (LDA) (Fisher, 1936), but unlike LDA is completely unsupervised.

In practice, we use an SGD-based optimizer to find a local optimum of $L(\theta)$. After computing the contrast features, we repeat the following process $T$ times. We first initialize a random direction $\theta$ with unit norm. Then for $E$ epochs, we calculate the loss according to Equation (1), and update the direction $\theta$ by via projected gradient descent, each time projecting $\theta$ back onto the unit sphere. Finally, we select the direction $\theta$ that has the lowest loss among the $T$ directions we found, and predict based on this direction. When we optimize the direction with multiple datasets, i.e. $S_1, ..., S_n$, we use the same algorithm, but average the loss across all datasets. In practice we use $T = E = 20$, and use Adam optimizer (Kingma & Ba, 2014) with a learning rate of 0.1.

## H STATISTICAL SIGNIFICANCE

Our main accuracy results for CCS and other methods (e.g. in Table 1 and elsewhere) are computed by evaluating the method on $40\%$ of the 1000 (or 500 in the case of COPA) examples sampled for each dataset, then averaging the resulting accuracy across 9 prompts per dataset (on average), 10 different datasets, and up to 5 models. This corresponds to about 180k samples in total; we performed subsampling in this way for computational efficiency reasons, as this is already substantial. For results where we average across datasets (i.e. most results in this paper), we average across 3800 IID samples (the original examples sampled from each dataset), then usually also averaged across a very large number of correlated samples: 9 prompts and (often) 5 different models for a given sample.

We can compute an upper bound on the standard error of accuracies computed in this way by ignoring the averaging across different prompts and models: this gives us a simple but coarse upper bound of the standard error of $\frac{1}{2\sqrt{3800}} \approx 0.8\%$. Applying a Wald test, the Wald statistic is $W = \frac{\mu_0 - \hat{\mu}}{se(\hat{\mu})^2} \geq \frac{\mu_0 - \hat{\mu}}{(0.008)^2}$, where (for example) $\mu_0$ is zero-shot accuracy and $\hat{\mu}$ is CCS accuracy. $W$ is then $\chi^2$-distributed with one degree of freedom.

We find that our main claims comparing accuracies are statistically significant at a 0.00001 level. This includes the claims that CCS outperforms zero-shot, with accuracies of 71.2% vs 67.2%, respectively;

that CCS is robust to misleading prompts while zero-shot isn't in the setting we test, with accuracies of 83.8% vs 70.9%, respectively; that CCS on MLM-pretrained DeBERTa substantially outperforms DeBERTa zero-shot, with accuracies of 93.7% vs 71.6%, respectively; and so on. Some minor observations are not necessarily statistically significant, such as that CCS (All Data) outperforms CCS on average (Section 3.2.1), but none of these are important for our main claims, and more powerful but complicated statistical tests would result in smaller p-values.

# I  DATASET SETUP

In this section, we describe the setup of our data and prompts in detail. We introduce all datasets we use and the way we convert them into a binary classification task. Our prompts derive from (Sanh et al., 2021)[4].

**Contrast Pair Example.** To illustrate how we construct contrast pairs, suppose we have a movie review "[text] = I loved this movie." and the sentiment of this review belongs to "[label0] = positive" or "[label1] = negative". We first format it into a binary question-answering or classification question: "[text] Is the sentiment of this example [label0] or [label1]? " using an existing zero-shot prompt. We then concatenate the question and candidate labels to create the contrast pairs:

$$x^+ = \text{[prefix] Q: Is the sentiment of "[text]" [label0] or [label1]? A: [label0]}$$

$$x^- = \text{[prefix] Q: Is the sentiment of "[text]" [label0] or [label1]? A: [label1]}$$

For instance, in this example, these would be:

$$x^+ = \text{Q: Is the sentiment of "I loved this movie." positive or negative? A: positive}$$

$$x^- = \text{Q: Is the sentiment of "I loved this movie." positive or negative? A: negative}$$

## I.1  TOKENIZATION

For all prompts we consider in this paper, we concatenate labels to the end of questions. The precise we do this depends on the model type. For encoder-decoder models such as T5, we replace line breaks (\n) with spaces, and we add a space between the question and proposed answer if the last token of questions is not already a space. For autoregressive models, we add a line break between the question and the proposed label if the last token of the question is neither a space nor a line break. For DeBERTa, we add a [SEP] split-token between the question and answer, and for RoBERTa, we add a  between the question and answer.

Finally, we add the prefix (either prefix in Figure 5, or the empty string for the regular zero-shot setting) to the beginning of the prompt, then compute the hidden representations for this pair to obtain the contrast features $\mathcal{C}$.

## I.2  DATASETS

We use ten datasets in our experiments. For each dataset, we select 1000 data points (except COPA, which has only 500 data points, so we only use those 500). When possible, we use just the test / validation split, but in other cases this is not enough data so we also use examples from the train split. For the most part this doesn't make a difference for our purposes because our methods are purely unsupervised.

It is important to note that T0 is an exception among the models, because but was trained on 9 out of the 10 datasets we consider (all except RTE). This may account for its anomalous performance. UnifiedQA was trained on BoolQ, which is one of the datasets we test on, but was not trained on any other evaluation set we consider, which we don't expect has an important qualitative effect.

We now enumerate the prompts we use for each dataset.

### I.2.1  AG_NEWS

AG_News is a topic classification dataset from (Zhang et al., 2015). Given a news paragraph, the label corresponds to the topic of this news, including "World politics", "Sports", "Business" and "Science and technology". For each example, we randomly select another label (which is incorrect), and randomly assign the correct label and this incorrect label to "label0" and "label1". With this, we construct the classification task using the following 8 prompts:

1  [prefix]Consider the following example: " ' [text] " ' Choice 1: [label0]. Choice 2: [label1].Between choice 1 and choice 2, the topic of this example is [label]

---

[4]https://github.com/bigscience-workshop/promptsource

2 [prefix]Consider the following example: "' [text] '" Choice 1: [label0]. Choice 2: [label1].Between choice 1 and choice 2, what is the topic of this example? [label]

3 [prefix]Consider the following example: "' [text] '" Which is the topic of this example, choice 1: [label0], or choice 2: [label1]? [label]

4 [prefix][text] Choice 1: [label0]. Choice 2: [label1].Between choice 1 and choice 2, the topic of this example is [label]

5 [prefix][text] Choice 1: [label0]. Choice 2: [label1].Between choice 1 and choice 2, what is the topic of this example? [label]

6 [prefix][text] Which is the topic of this example, choice 1: [label0], or choice 2: [label1]? [label]

7 [prefix][text] What label best describes this news article, choice 1: [label0], or choice 2: [label1]? [label]

8 [prefix][text] Which section of a newspaper would this article likely appear in, choice 1: [label0], or choice 2: [label1]? [label]

Here the last "[label]" is "choice 1" for $x^+$ and "choice 2" for $x^-$. Notice that we only specify "[label]" for the prompts that are constructed manually. For prompts in (Sanh et al., 2021), we leave their labels unchanged. (For example, we use "choice 1" or "choice 2" but they can use "Animal" or "Plant")

### I.2.2   AMAZON_POLARITY

Amazon_polarity is a sentiment classification task from (McAuley & Leskovec, 2013). The content is the review of goods in Amazon, and the label can be "negative" or "positive". We use 11 different prompts in this dataset. We first take all prompts from (Sanh et al., 2021) (Page 164, 9 prompts in total), and the two of our own as follows:

1 [prefix]Consider the following example: "' [content] '" Between [label0] and [label1], the sentiment of this example is [label]

2 [prefix]Consider the following example: "' [content] '" Between [label0] and [label1], which is the sentiment of this example? [label]

Here "[label]" is "negative" for $x^+$ and "positive" for $x^-$.

### I.2.3   BOOLQ

BOOLQ is a QA task where each example consists a yes/no question from (Clark et al., 2019). We directly use the 10 prompts from (Sanh et al., 2021) (Page 146)

### I.2.4   COPA

COPA is a causal reasoning task to determine either the cause or the effect of a given premise (Roemmele et al., 2011). Here the label is a short sentence. We use 10 prompts, where 9 are from (Sanh et al., 2021) (Page 177), and we add one more prompt:

1 [prefix]Consider the following premise: "' [premise] '" Choice 1: [choice1] Choice 2: [choice2] Q: Which one is more likely to be the [question], choice 1 or choice 2? [label]

### I.2.5   DBPEDIA_14

DBpedia_14 is a topic classification dataset constructed by picking 14 non-overlapping classes from DBpedia 2014 (Lehmann et al., 2015). We manually create 8 prompts. For each example, we randomly select the incorrect label from the remaining 13 classes, and randomly assign the correct label and this incorrect label to "[label0]" and "[label1]".

1 [prefix]Consider the following example: "' [content] '" Choice 1: [label0]. Choice 2: [label1].Between choice 1 and choice 2, the topic of this example is [label]

2 [prefix]Consider the following example: "' [content] '" Choice 1: [label0]. Choice 2: [label1].Between choice 1 and choice 2, what is the topic of this example? [label]

3 [prefix]Consider the following example: "' [content] '" Which is the topic of this example, choice 1: [label0], or choice 2: [label1]? [label]

4 [prefix][content] Choice 1: [label0]. Choice 2: [label1].Between choice 1 and choice 2, the topic of this example is [label]

5 [prefix][content] Choice 1: [label0]. Choice 2: [label1].Between choice 1 and choice 2, what is the topic of this example? [label]

6 [prefix][content] Which is the topic of this example, choice 1: [label0], or choice 2: [label1]? [label]

7 [prefix][content] What category does the paragraph belong to, choice 1: [label0], or choice 2: [label1]? [label]

8 [prefix][content] What label best describes this paragraph, choice 1: [label0], or choice 2: [label1]? [label]

Here "[label]" is "choice 1" for $x^+$ and "choice 2" for $x^-$.

### I.2.6 IMDB

IMDB is a sentiment dataset from (Maas et al., 2011). Given a movie review, the label is either "[label0] = negative" or "[label1] = positive. We use 13 prompts, where 11 are from (Sanh et al., 2021) (Page 168), and the rest two are as follows:

1 [prefix]Consider the following example: "' [text] '" Between [label0] and [label1], the sentiment of this example is [label]

2 [prefix]Consider the following example: "' [text] '" Between [label0] and [label1], which is the sentiment of this example? [label]

Here "[label]" is "negative" for $x^+$ and "positive" for $x^-$.

### I.2.7 PIQA

The PIQA dataset measures the physical commonsense reasoning ability of models. We use 11 prompts for PIQA, all of which are from (Sanh et al., 2021)(Page 160). The label is a complete sentence that can be the solution of the question.

### I.2.8 QNLI

QNLI from (Rajpurkar et al., 2016) is a question-answering dataset consisting of question-paragraph pairs, where one of the sentences in the paragraph (drawn from Wikipedia) contains the answer to the corresponding question (written by an annotator). We use 5 prompts from (Sanh et al., 2021). The label is either "yes" or "no" depending on whether the information in the paragraph is enough to answer the paragraph.

### I.2.9 RTE

RTE is a textual entailment dataset (Wang et al., 2018). The label corresponds to whether the text entails the hypotheses. We use 11 prompts, where 10 are from (Sanh et al., 2021)(Page 48), and where we manually add one more prompt:

1 [prefix][premise] Question: Does this imply that "[hypothesis]", yes or no? [label]

Here "[label]" is "yes" for $x^+$ and "no" for $x^-$.

### I.2.10 STORY_CLOZE

Story Cloze is a story completion task from (Mostafazadeh et al., 2017). Given a short story, the task is to determine which of two endings is more likely to continue that story. We use 9 prompts, where 6 are from (Sanh et al., 2021), and the remaining 3 are as follows:

1 [prefix]Consider the following story: "' [input_sentence_1] [input_sentence_2] [input_sentence_3] [input_sentence_4] '" Choice 1: [sentence_quiz1] Choice 2: [sentence_quiz2] Which is the more plausible ending of this story, choice 1 or choice 2? [label]

2 [prefix]Consider the following story: "' [input_sentence_1] [input_sentence_2] [input_sentence_3] [input_sentence_4] '" Choice 1: [sentence_quiz1] Choice 2: [sentence_quiz2] Which is the more plausible ending of this story? [label]

3 [prefix][input_sentence_1] [input_sentence_2] [input_sentence_3] [input_sentence_4] Choice 1: [sentence_quiz1] Choice 2: [sentence_quiz2] Which is the more plausible ending of this story, choice 1 or choice 2? [label]

Here "[label]" is "choice 1" for $x^+$ and "choice 2" for $x^-$.

