# OpenReview forum: "Discovering Latent Knowledge in Language Models Without Supervision"
_ICLR.cc/2023/Conference — ICLR 2023 poster_

### Official Review · Reviewer_32x3 · 2022-10-22

**Confidence:** 4
**Correctness:** 2
**Technical Novelty And Significance:** 3
**Empirical Novelty And Significance:** 4
**Recommendation:** 6

**Clarity, Quality, Novelty And Reproducibility:**

Clarity: In general, this paper is well-written and organized. There are several parts that confuse me:
- Why do you use just 1000 examples per dataset? Can you give a justification for this design choice?
- In table 3, how do you add the contrastive prompts to the encoder-only and encoder-decoder models? To me, these models are not designed to work with prepended prompts.

Novelty: To my knowledge, the approach is novel and intriguing and I liked the idea.

Quality: The experiments are comprehensive and have extensive discussions about the results. However, I don't think all of the claims are well supported. See weaknesses for details.

Reproducibility: The results should be reproducible by using the code provided.

**Strength And Weaknesses:**

Strengths:
- The proposed approach is novel to my knowledge.
- Strong empirical results across many datasets and models.
- The problem this paper trying to tackle is fundamental

Weaknesses:
- My major concern about this paper is that the claims made are not well supported by the experiments.
	- In section 3.2.2, "Recall our motivating goal: to discover latent knowledge in a language model even when its training objective causes the model to output false text." I don't see how this claim is related to the experiments in this section. The experiment is conducted in a zero-shot setting in which no training objectives are used. What they really do is use a set of contrastive prompts to mislead the model and the experimental results show that **CCS can improve the robustness of the model**. That's the takeaway I can conclude from the experiment.
	- In section 3.3.1, "CCS FINDS A TASK-AGNOSTIC REPRESENTATION OF TRUTH". I am not super convinced by this claim, as all of the models (without fine-tuning using the CCS objective) used for this experiment can already make zero-shot generalizations on those datasets to some extent. It is unclear to me what is added by the CCS objective. Does it just make the model more robust against domain shifts? Or it captures something else? Moreover, do you have a more formal definition of TRUTH here?
	- In section 3.3.2 "CCS CAN FIND INTERMEDIATE REPRESENTATIONS OF TRUTH". I don't quite get the logic behind this experiment. How can we conclude by looking at the results that lower layers perform worse? Can you elaborate on this a bit more?
- The baseline seems a bit weak to me. How about you just fine-tune the models used in the paper using the original training objective on those datasets used for testing?
- The "purely unsupervised" setting seems to be a bit overclaiming. In the experiments, they still need a non-trivial number of training examples to make the proposed approach work. It would make the paper much stronger if they could use examples from a more "wild" setting, e.g., sentences from a random corpus, to train CCS and show it could still work.


**Summary Of The Paper:**

This paper proposed a novel approach to address a fundamental problem in supervised machine learning: misalignments exist between a training objective and the truth. Instead of telling the model "what is true" by human annotations, they proposed to learn the truth by the model itself in a purely unsupervised way. Specifically, they introduced Contrast-Consistent Search (CCS), a method that learns a linear projection of the hidden states that satisfies logical consistency (consistent across negations). Given a yes-no question, they first construct two instances with the binary answers concatenated to the question. They then train a classifier to predict the binary labels using the hidden states of a pretrained language model. To train the classifier, they add several constraints to insure logical consistency. For example, a statement and its negation should have probabilities that add up to 1. They conducted experiments using 6 different models and the empirical results on 10 question answering datasets showed they can improve the zero-shot performance by 4% compared to the baselines.

**Summary Of The Review:**

In general, the proposed approach is novel and the empirical results seem strong. However, at the current stage, several claims are not well supported by the experiments and it is not super clear what the takeaways are. At the current stage, I lean toward rejection but I think improvements can be made to make it a very strong publication.

---

> ### Author Response · Authors · 2022-11-14
> **Response Part 1: Clarifying Our Motivation Behind Sections 3.2.2 and 3.3.1**
>
> Thank you very much for your review. We are glad that you believe our “proposed approach is novel”, that you thought we had “strong empirical results across many datasets and models”, and that you agree “the problem this paper [is] trying to tackle is fundamental”.
>
> **Evidence for “CCS Is Robust To Misleading Prompts”**
>
> Your first main concern is that you’re not sure how our results in Section 3.2.2 (with misleading prompts) is related to our goal to “discover latent knowledge in a language model even when its training objective causes the model to output false text” since there are no training objectives used in this setting.
>
> To clarify, here is the relationship between this experiment and the training objective:
>
>
>
> * The standard language modeling objective is to predict what the next token would be given previous tokens.
> * Our claim is that language models sometimes output false text *because of this training objective* A common way this happens is through context imitation: if it is prompted with a context that is flippant or erroneous, then new outputs are more likely to be flippant or erroneous as well.
> * Our goal with the experiment in Section 3.2.2 was to elicit this context-following effect by prompting models with examples of incorrectly answered questions; if the model does a sufficiently good job of predicting the next token (its original training objective), it should output incorrect answers with this prompt, even if it is capable of outputting correct answers, due to imitating the context (which is incentivized by the next-token objective, since the context usually gives useful clues for how the sentence should be completed).
> * In contrast, if we have a method for recovering a model’s internal “knowledge”, it should still work well even when its outputs are misled in this way; we ran the misleading prompt experiments as an initial sanity check that this indeed holds.
>
> We agree that there are subtleties in the interpretation of this result – indeed, that is why we included footnote on page 6, and that is why we emphasize that it is “an initial proof of concept” for this long-term goal in the second sentence of 3.2.2. Nevertheless, despite these caveats, we think it is a suggestive result.
>
> We have revised 3.2.2 paper to clarify the connection between these experiments and this goal. Please let us know if you still find anything about this confusing.
>
> **Evidence for “CCS Finds a Task-Agnostic Representation of Truth”**
>
> Your second main concern seems to be that all of the models can already generalize in a task-agnostic way using zero-shot prompting, so it is unclear what is added by CCS.
>
> We agree that the default zero-shot responses from language models are also task-agnostic; the problem with zero-shot isn't that it is task-specific, but that it does not necessarily capture what a model actually knows.
>
> The purpose of this section is simply to rule out a possible objection one might have: the possibility that *unlike* zero-shot, our method might *not* be task-agnostic. We tried to explain this in the first two sentences of 3.3.1:
>
> _“From the results described so far, it may be possible that the classifier we find is capturing dataset-specific properties, such as which of two labels (e.g. ``Yes`` vs. ``No``) is more likely to be correct. We rule this out by showing that it generalizes across completely different tasks, including ones with different label spaces (such as from ``Yes`` and ``No`` for a generic task to ``Positive`` and ``Negative`` for sentiment classification).”_
>
> If this still isn’t clear, please let us know and we can modify the language to clarify what we meant.

---

> > ### Comment · Reviewer_32x3 · 2022-11-17
> > **My concerns are not fully addressed**
> >
> > Thanks for the clarifications.
> >
> > Re: Evidence for “CCS Is Robust To Misleading Prompts”
> >
> > As I commented in my original review, I see CCS is robust to misleading prompts but this does not entail CCS is able to recover a model's "internal knowledge". For example, if we fine-tune the model using some adversarial examples, the model will also be more robust to misleading prompts. However, we cannot say the model recovers a model's "internal knowledge" through adversarial training. BTW, how do you define "internal knowledge" here?
> >
> > Re: Evidence for “CCS Finds a Task-Agnostic Representation of Truth”
> >
> > Thanks for the clarification; it makes sense to me. However, I would still like to see what is added by the CCS objective compared to a raw language model.

---

> > > ### Author Response · Authors · 2022-11-18
> > > **Followup Response (Part 1)**
> > >
> > > Thank you very much for your prompt response!
> > >
> > > **As I commented in my original review, I see CCS is robust to misleading prompts but this does not entail CCS is able to recover a model's "internal knowledge"... how do you define "internal knowledge" here?**
> > >
> > > To clarify, when we say that CCS can recover a model’s “internal knowledge”, we simply mean that CCS is _not just recovering knowledge in the model outputs_. We provide several pieces of evidence for this claim – only one of which is that CCS is robust to misleading prompts – in our revised Section 3.3.2, which also includes a couple new experiments (see below for a summary of these new experiments), and which we recommend reading if you haven’t already.
> > >
> > > Moreover, in the context of the misleading prompts experiments, our main piece of evidence that CCS is recovering “internal knowledge” (in the sense defined above) is *not* that CCS is robust. It is instead that CCS is robust *but model outputs are not*. In other words, simply want to show the *gap* between CCS using the internal model activations and zero-shot using the model outputs.
> > >
> > > **I would still like to see what is added by the CCS objective compared to a raw language model.**
> > >
> > > We believe our claim for this is also likely best summarized by our revised section 3.3.2. Some of the advantages of CCS relative to raw language models include:
> > > - Empirically our approach does better than using a raw language model’s outputs (as measured by accuracy on a large number of tasks, see Section 3.2.1).
> > > - Language models are simply trained to imitate human text, which is not necessarily correct, so we should expect them to output false text even if they “know better”, which bounds the quality of the knowledge we can probe using model outputs (as we describe in the Introduction).
> > > - CCS is *much more robust* to the misleading prompt we try in our experiments than raw language model outputs (Section 3.2.2)
> > > - In our revised paper, we restructured Section 3.3.2 and added a couple experiments to more clearly show how our approach does not just recover the knowledge in the model outputs. In particular, we show that CCS can work in settings where model outputs are not very informative, and we show that using intermediate activations (rather than last-layer activations or model outputs) can be crucial for achieving high accuracy.

---

> ### Author Response · Authors · 2022-11-14
> **Response Part 2: Clarifying The Logic Behind 3.3.2, Our Choice of Baselines, and Why Our Approach Really is "Purely Unsupervised"**
>
> **What is the logic behind “CCS Can Find Intermediate Representations of Truth”?**
>
> Another reviewer also had confusions about 3.3.2, so we decided to restructure this section to clarify our argument and to add some additional results. The main goal of this section is to argue that CCS can recover knowledge that is different from – and sometimes more useful than – a model’s outputs.
>
> In the revised version of this section, we show this in several ways:
> 1. We argue that previous results in the paper, which showed that CCS outperforms zero-shot in several settings, already provides strong evidence for this claim.
> 2. We add an additional result showing that while CCS with the intermediate-layer activations are robust to misleading prompts (3.2.2), CCS with the last-layer activations are *not* robust to misleading prefixes. This shows that using intermediate activations can be more decorrelated from – and more useful – than the model outputs.
> 3. We add an additional result showing that CCS can work in settings where model outputs are not informative – namely, when we use masked language models without any [MASK] tokens, and when we construct contrast pairs with the labels in the middle of a prompt rather than at the end. This also shows that CCS can recover useful knowledge from model activations even when model outputs aren’t very useful.
>
> Please let us know if our revised Section 3.3.2 is more clear to you, or if there is anything else you are still confused by that we can address. Thank you.
>
> **The baseline seems a bit weak**
>
> You were also wondering why we don’t just fine-tune models as a stronger baseline.
>
> First, note that this is somewhat close to what our logistic regression (LR) ceiling does: it learns a linear probe on the activations in a supervised way. However, this – and finetuning of the entire network, which you seem to be suggesting – are both inappropriate baselines for the setting we consider, since they both require labels. Since we focus on the purely unsupervised setting, we treat LR as a ceiling rather than as a baseline, and instead primarily focus on strong zero-shot baselines.
>
> **The “purely unsupervised” setting seems to be a bit overclaiming… they still need a non-trivial number of training examples to make the proposed approach work**
>
> First, to clarify we do not actually need many training examples to make the proposed method work; Section 3.3.3 shows that it can often do well with only a handful of points (e.g. 16 or fewer). Moreover, even if it did require a large number of examples, Section 3.3.1 shows that it can transfer across different tasks, so we could just collect training examples from one domain and apply it to new samples in a different domain without retraining at test time. Finally, we would maintain that our method really is “purely unsupervised” – we do not see how that is overclaiming.
>
> **Conclusion**
>
> We believe our responses address all of your main concerns; if they do, we would appreciate it if you raised your score accordingly. If not, please let us know so that we can address any serious concerns that you might still have.
>
> Thank you very much!

---

> > ### Comment · Reviewer_32x3 · 2022-11-17
> > **My concerns are not fully addressed**
> >
> > Thanks for the clarifications!
> >
> > Re: The baseline seems a bit weak
> >
> > Sorry for making the confusion in my original review. By fine-tuning the model using its original training objectives, I mean the language model objective like Masked language modeling (MLM). You don't necessarily need labels here in my opinion: you can simply use the same data you used to train CCS and feed them to other models. For encoder-decoder models, you can simply make two training instances with the same input but different outputs (one positive and one negative). For encoder-only and decoder-only models, you can fine-tune the models using the original objectives on your constructed datasets containing both positive and negative examples.
> >
> > Re: The “purely unsupervised” setting seems to be a bit overclaiming
> >
> > I tend to agree with your opinion but it would be better to have an experiment showing how many examples you need to outperform the base model.

---

> > > ### Author Response · Authors · 2022-11-18
> > > **Followup Response (Part 2)**
> > >
> > > **By fine-tuning the model… you can simply use the same data you used to train CCS and feed them to other models**
> > >
> > > Thank you for your clarification; we believe we understand your question much better now.
> > >
> > > However, we believe this proposal would lead to a degenerate solution: if you construct contrast pairs and train a model using a LM or MLM objective on those pairs (e.g. training it to predict “Is 2+2+4? Yes” and training it to predict “Is 2+2=4? No”), then it should learn to predict the two possible labels (e.g. “Yes” and “No”) to each have 50% probability, which should make model outputs actually uninformative, and we are not sure how one could modify your proposal to fix this.
> > >
> > > **it would be better to have an experiment showing how many examples you need to outperform the base model**
> > >
> > > To clarify, we believe our results looking at performance as a number of examples (Figure 3) provides this information when combined with the zero-shot accuracy numbers in Table 1. Please let us know if there’s anything else you’re looking for not shown there.
> > >
> > > Do you have any remaining major concerns that we can address, or do you think the paper should now be a “weak accept”?
> > >
> > > Thank you again for your quick response!

---

> > > > ### Comment · Reviewer_32x3 · 2022-11-19
> > > > **Raise my score to "weak accept"**
> > > >
> > > > Thanks for the detailed explanation. I agree there may not be a trivia way to make the simple baseline work except for the encoder-only models. Most of my major concerns are resolved and I would like to raise my score to "weak accept". Will discuss more during the final reviewer discussion stage.

---

### Official Review · Reviewer_eJhP · 2022-10-24

**Confidence:** 3
**Correctness:** 3
**Technical Novelty And Significance:** 3
**Empirical Novelty And Significance:** 3
**Recommendation:** 6

**Clarity, Quality, Novelty And Reproducibility:**

Clarity: The paper reads clearly and is well-written.

Quality: The experiments and analysis seem of high quality.

Novelty: The CCS probe seems to be a novel way to probe for knowledge by posing knowledge as a binary classification task.

Reproducibility: Code is provided but I have not independent verified if it is runnable / reproducible.

**Strength And Weaknesses:**

Strengths:
- The paper targets an important issue with language models: verifying truthfulness of outputs and whether knowledge is learned within the model parameters.
- There is an interesting result in 3.2.2 for how CCS is robust to adversarial prompts. I would have liked to see more discussion on why this particular design of "misleading prefix" was chosen, however.
- CCS is demonstrated to be usable with multiple models across a wide range of tasks.
- Good discussion of limitations and possible future challenges.

Weaknesses:
I'm having a bit of a challenge reconciling the challenge of verifying truthfulness with the focus on results of how CCS out-performs zero shot approaches on accuracy - as mentioned in the paper, it *does* indicate that there is a disconnect between inferences made via one modality (ranking of activations) vs. direct model outputs, but it is difficult to see how that shows that one method is "better" than the other for probing knowledge.

**Summary Of The Paper:**

The authors introduce an unsupervised probe for knowledge in large language models to tackle a perceived challenge in verifying truthfulness of predictions/generated text. The authors' proposed method involves classifying/ranking model activations between binary labels/framings of a yes-no question. The authors demonstrate that their CCS probe is resilient to prompt-based attacks.

**Summary Of The Review:**

This paper tackles an important challenge with verifying truth and knowledge in language models, but I have some questions about the basic premise of how it shows what it claims (a better way to probe for knowledge).

---

> ### Author Response · Authors · 2022-11-14
> **Clarifying The Advantages of Using Activations Over Zero-Shot**
>
> Thank you for your positive review! We are glad you liked our paper.
>
> **Why is probing via activations better than probing via outputs?**
>
> It seems like your main concern is that even if probing knowledge from models via their activations is different from probing knowledge via their outputs, it’s not clear why the former is preferable to the latter.
>
> There are several reasons why activations may be preferable:
> * Empirically our approach does better than using model outputs (as measured by accuracy on a large number of tasks, see Section 3.2.1).
> * Language models are simply trained to imitate human text, which is not necessarily correct, so we should expect them to output false text even if they “know better”, which bounds the quality of the knowledge we can probe using model outputs, as we describe in the Introduction.
> * In our revised paper, we restructured Section 3.3.2 and added a couple experiments to more clearly show how our approach does not just recover the knowledge in the model outputs. In particular, we show that CCS can work in settings where model outputs are not very informative, and we show that using intermediate activations (rather than last-layer activations or model outputs) can be crucial for achieving high accuracy.
>
> Of course, our method may not be the optimal way of probing knowledge from activations in an unsupervised way. However, our results show that it already has important advantages over zero-shot probing. We are excited about this as an alternative paradigm for probing what models know that may work even when model outputs are unreliable or unavailable.
>
> Please let us know if you still have any questions or concerns about this (or any other) point!

---

> ### Author Response · Authors · 2022-11-18
> **Gentle Reminder To Let Us Know If You Have Any Remaining Questions**
>
> Thank you again for your positive review of our paper! We just wanted to check in one last time before the deadline tomorrow: do you have any remaining questions that we can address? If not, and you think our response addressed what you viewed as the main weakness of our paper, do you think our paper should now be a clear "accept"?
>
> Thank you!

---

### Official Review · Reviewer_r2Hq · 2022-10-24

**Confidence:** 3
**Correctness:** 3
**Technical Novelty And Significance:** 4
**Empirical Novelty And Significance:** 4
**Recommendation:** 6

**Clarity, Quality, Novelty And Reproducibility:**

The paper is of generally high-quality writing and the method and insights appear to be novel.
Degree of reproducibility appears to be high from the textual description and supplementary material.

Regarding clarity, there are some questions/suggestions:
* Why do you subsample only 1000 examples from each dataset? Should be justified in the text.
* What do you mean by "we optimize 10 times using AdamW"? Are these 10 different seeds? If so, what is the impact of seed tuning?
* What is the added benefit of Figure 2 in comparison with the average statistic? It does not appear very helpful to support the argument, which already pretty strong (for this one model).
* Figure 3: The absolute task performance (alongside its color coding) is not very meaningful with respect to what you want to show, which is that transfering from one task to another is doable. The figure is therefore not easy to grasp. I suggest to change figure 3 such that each cell represents the difference in performance with respect to "No Transfer". That way the reader would immediately understand from the colors that transfering almost never degrades the performance.

**Strength And Weaknesses:**

Strengths:
* very strong motivation
* simple and effective idea
* very valuable empirical results to the community
* discusses relevant alternative hypotheses

Weaknesses:
* **update post rebuttal**: my biggest concerns have been addressed
* statistical validity is unclear due to small test sets
    * why are the datasets subsampled to include only 1000 examples, of which 400 are the test set?
    * these are unneccessarily small test sets, causing greater uncertainty and therefore confidence in the results
    * can you please justify why you believe that differences are unlikely to be due to randomness? Note that stds in Table 1 are relatively large.
* evidence for disproving alternative hypotheses is weak in several cases:
    * Section 3.2.2: CCS Is Robust To Misleading Prompts
        * goal: "discover latent knowledge in an LM even when its **training objective** causes the model to output false text"
        * method: change **test time** prompts and observe difference
        * method is not well suited for the goal, because you don't touch the training objective of the model at all.
        * result is found for only one out of 5 models. Combined with the statistical uncertainty above, I am not convinced that this is sufficient evidence that this is a valid, general finding.
    * Section 3.3.2: CCS Can Find Intermediate Representations of Truth
        * motivation: output text might encode truth, and CCS may merely correlate with output text
        * method: show that CCS applied to intermediate layers also works well.
        * implicit assumption: lower layers don't correlate with the produced output text (as much?). This needs evidence of some kind, e.g. a reference.
        * results: lower layers perform worse, but according to the paper, they still perform well enough. It isn't clear what constitutes well enough.
        * (their) conclusion: lower layers do well despite not correlating with the output text, so they have to encode truth qualitatively different from the output text.
        * Lower performance at lower layers may be entirely explained by decreasing (but not vanishing) correlation with the output text. Since there is no quantification of the latter, we cannot make the above conclusion.

Alternative method to prove that internal truth values are present independent of the input:
* if truth values are encoded independent of textual output, you should be able to extract them by:
    1. replacing positive and negative labels with completely uninformative, random text. Since you aim to normalize out the effect of "Yes" and "No", the appended text shouldn't matter either way, or
    2. don't alter x_i, but train the model with two different classification heads.

**Summary Of The Paper:**

The paper aims to understand language models' internal knowledge about truth values of the text that it outputs.
Given a set of yes-no questions, they train a truth classifier in an unsupervised way by first extracting the hidden states of the language model when it gives a positive or negative answer, respectively. They train the classifier to be consistent, i.e., the probabilities of the two outcomes should sum to one, and confident (one label should receive high probability).
The model is tested on zero-shot binary classification on a large set of tasks. It performs reasonably well, outperforming standard LM-based zero-shot classification on average by a substantial margin. They further discuss and test several alternative hypotheses for their model's success, aiming to show that their model indeed captures internal information that is qualitatively different from the information in the model's textual output.

**Summary Of The Review:**

I applaud the technical and empirical novelty of the paper. I think it will be of significant interest to many in the community. The main claims are fairly well supported. As noted in the rebuttal, some aspects of the methodology are confusing and could maybe be executed and/or explained/justified better.

---

> ### Author Response · Authors · 2022-11-14
> **Response Part 1: Our Main Results Are Actually Averaged Across Hundreds of Thousands of Examples**
>
> Thank you very much for your detailed review. We are glad that you liked many aspects of our paper (thinking that it has a “very strong motivation”, that it is a “simple and effective idea”, and that it has “very valuable empirical results to the community”, etc.).
>
> **Statistical Validity**
>
> One of your main concerns is about whether our main results (especially in Table 1) are statistically valid since we select only 1000 examples from each dataset and use only 400 of those examples as the corresponding test set, especially given that the standard deviations in Table 1 are so high.
>
> We believe this concern may be stemming from a misunderstanding about what the numbers in Table 1 correspond to. The standard deviations in Table 1 do _not_ correspond to the standard error in estimating the accuracy; it is instead the average standard deviation _in accuracy_ _across different prompts_. It is well-known that language models are very sensitive to different prompts, so we would expect this standard deviation to be high. We modified the writing to make this point clearer.
>
> Moreover, our main _accuracy_ results in Table 1 are averaged over all 10 datasets and all ~9 prompts. This means that for each model we average over approximately 400*9*10 = 36k examples, and when averaging over all models (the right-most column of Table 1) we average over approximately 36k*5 = 180k examples, which is already much larger than many datasets. We tweaked the language to make this clearer.
>
> The reason we decided to take a subset of 1000 examples from each dataset is because this actually corresponds to ~180k examples, which is already quite large. Moreover, computing the hidden states of large language models can require very small batch sizes to fit in memory even when parallelized across several GPUs – e.g. T5 -based models (3 out of our 6 models) require a batch size of one even when parallelized across multiple 40GB A100 GPUs – so evaluating on the full test sets of every dataset we consider would have been computationally prohibitive. For example, evaluating on the full Amazon sentiment test set alone (just one of ten datasets) would have required computing the hidden states of about thirteen million examples (200k test examples * 11 prompts for Amazon * 6 models), which we estimate would have required 1-2 weeks to compute using our hardware.
>
> We have updated our paper to add these clarifications. Please let us know if you have any other concerns about statistical validity that we can address.
>
> **Evidence for “CCS Is Robust To Misleading Prompts”**
>
> Your next main concern is about whether our method of “observe changes in accuracy when we change the prompts” is well-suited to the goal of “discovering latent knowledge in a LM even when its training objective causes the model to output false text”.
>
> Thank you for raising this concern. We tried to explain this point in the paper, but we realized our original phrasing was confusing. We had previously said (in the first paragraph of 3.2.2): “we add a prefix to the beginning of our zero-shot prompts that consists of questions answered incorrectly (Figure 7). The hope is that such a prefix will **decrease zero-shot accuracy because the model will imitate its context** by outputting incorrect answers”.
>
> To flesh this out more, here is the relationship between this experiment and the training objective:
>
> * The standard language modeling objective is to predict what the next token would be given previous tokens.
> * Our claim is that language models sometimes output false text *because of this training objective* A common way this happens is through context imitation: if it is prompted with a context that is flippant or erroneous, then new outputs are more likely to be flippant or erroneous as well.
> * Our goal with the experiment in Section 3.2.2 was to elicit this context-following effect by prompting models with examples of incorrectly answered questions; if the model does a sufficiently good job of predicting the next token (its original training objective), it should output incorrect answers with this prompt, even if it is capable of outputting correct answers, due to imitating the context (which is incentivized by the next-token objective, since the context usually gives useful clues for how the sentence should be completed).
> * In contrast, if we have a method for recovering a model’s internal “knowledge”, it should still work well even when its outputs are misled in this way; we ran the misleading prompt experiments as an initial sanity check that this indeed holds.
>
> We have revised 3.2.2 paper to clarify the connection between these experiments and this goal. Please let us know if it is still confusing to you.

---

> > ### Comment · Reviewer_r2Hq · 2022-11-17
> > **Reply to response**
> >
> > Thank you for your clarifications.
> >
> > **Statistical Validity**
> > I understand that invoking these large models is expensive, and that you're doing a lot of forward passes already. My concern regarding statistical validity remains, however. Comparing averages over different tasks with potentially different scales is already sketchy. I don't know how it is with averaging over prompts and models. Especially combined with small datasets, I can not intuitively be confident that the observed differences in performance are actually meaningful (even though I like your model and would like to believe its results!). What statistical test could you apply in the specific case of your paper? These considerations should probably be discussed more in the paper.
> >
> > **Evidence for “CCS Is Robust To Misleading Prompts”**
> > Thank you for your clarifications. I think your argumentation is reasonable, but the results are still rather weakly convincing. I understand it's a proof of concept, but a proof of concept is probably just not strong enough evidence for such a prominent part of the paper.
> >
> > **Decorrelating CCS from the output and alternative method to find truth values**
> >
> > Your paper has two concepts that are, at least in my mind, conflicting or at least confusing. On the one hand, you want CCS to be decorrelated from the output of the model, so you can essentially identify whether it is lying. On the other hand, your method is based on generating contrastive pairs that differ **in their output** (which you then normalize out again, adding to the confusion).
> >
> > Assume you have an arbitrary statement $x$ (e.g., cats are mammals). How can you use your method to find the truth value of this statement?
> > My suggestion is to use two different classification heads $p_+$ and $p_-$, that you both apply to $x$ and train them with the same consistency and confidence loss as before.
> >
> > **Evidence for “CCS Can Find intermediate Representations of Truth”**
> >
> > I think your additional experiments are good and somewhat convincing.

---

> > > ### Author Response · Authors · 2022-11-18
> > > **Followup Response**
> > >
> > > Thank you very much for your prompt reply!
> > >
> > > **Comparing averages over different tasks with potentially different scales is already sketchy.**
> > >
> > > To clarify, all of the tasks are converted to binary QA tasks, so the scales are the same: the metric is always accuracy, where 50% is random chance and 100% is perfect accuracy. Of course, different tasks have different levels of difficulty. However, averaging accuracy across different tasks is common practice, so we don’t see any issue with that. But please let us know if we misunderstood something.
> > >
> > > **What statistical test could you apply in the specific case of your paper?**
> > >
> > > To address your concern, we use Wald’s test to test the statistical significance of our main results. First, note that for each accuracy number in Table 1, we average over at least 3800 IID examples (i.e. the original samples from all datasets). We also average over a large number of prompts and (in some cases) several models, which can have complicated correlations in general. However, even taking a worst-case view of these correlations, the standard error of our main reported accuracies are *upper bounded* by 1/(2*sqrt(3800)) ~= 0.8%. Applying a Wald test with this upper bound, our main claims are statistically significant at a 0.00001 level. This includes the claims that CCS outperforms zero-shot, with accuracies of 71.2% vs 67.2%, respectively; that CCS is robust to misleading prompts while zero-shot isn’t in the setting we test, with accuracies of 83.8% vs 70.9%, respectively; that CCS on MLM-pretrained DeBERTa substantially outperforms DeBERTa zero-shot, with accuracies of 93.7% vs 71.6%, respectively; and so on. Some minor observations, such as that CCS with All Data outperforms standard CCS on average, are not revealed to be statistically significant using this upper bound, but none of these observations are important for our main claims. Moreover, Wald’s test is fairly crude and more powerful tests such as a paired t-test would give smaller p-values; we would be happy to do this if it is a sticking point.
> > >
> > > We added a discussion of this to the paper (Appendix I) because of your suggestion. Are there any remaining major claims of ours where you are concerned about statistical significance (so that we can potentially run another statistical test to show that it’s not an issue)?
> > >
> > > **Evidence for “CCS Is Robust To Misleading Prompts”**
> > >
> > > You were also concerned the misleading prompt section is too prominent in the paper. In part also thanks to another reviewer’s suggestion, we decided to move Figure 4 (the plot we had for the misleading prompts subsubsection) to the appendix, so that this section is now only a little more than a paragraph, de-emphasizing it slightly.
> > >
> > > **Decorrelating CCS from the output and alternative method to find truth values**
> > > Thank you for the clarification; we believe we understand your confusion and your proposal better now.
> > >
> > > First, regarding your confusion: to clarify, the essential property we require is that we can generate contrast pairs that differ *in their input* rather than in their *output*. In the case of generative models like GPT-J the inputs are closely related to the outputs (since at each token location t the output at location t is the prediction for what the (t+1)st token will be). However, our method works more generally than that. For instance, MNLI-finetuned RoBERTa and DeBERTa are classification models that don’t have any natural language outputs, and when we use the encoder activations of encoder-decoder models we “throw out” the decoder half of the model that connects inputs to natural language outputs.
> > >
> > > Second, we believe your clarified proposal leads to a degenerate solution, and we are not sure how it can be modified to avoid this issue. In particular, if we only train $p_+$ and $p_-$ as heads to satisfy the confidence and consistency objective on a statement x (like “cats are mammals”), then it can get perfect loss by always setting $p_+ = 1$ and $p_- = 0$. We are not sure how to get around this because it seems essential for our method that we have some way of counterfactually changing the truth value of an input.
> > >
> > > Please let us know if you have any other questions or concerns about this or anything else.

---

> > > > ### Comment · Reviewer_r2Hq · 2022-11-18
> > > > **Follow up response to follow up response to**
> > > >
> > > > Your discussions around statistical validity made me more confident about this work.
> > > >
> > > > Regarding the two classification heads, I agree that it can easily lead to a degenerate solution. Adding an entropy term might help here, no? And the degenerate solution can also appear for your method, right? $p(x^+) = 1, p(x^-)=0$ should do the trick. I guess the reason why this doesn't happen is that you normalize out the positive and negative labels...?
> > > > If still possible, could you run an ablation without the normalization to confirm that this indeed is the case?
> > > >
> > > > Either way, I think your paper doesn't have any major flaws anymore. Your rebuttal with all the additional information was very helpful.
> > > > I am still fuzzy about the whole input/output discussion. I think there is a 50% chance that something just doesn't click on my end, but maybe it could also be explained better in the paper. I think adding contrastive text pairs is not very elegant, because you need to worry about the formatting and all that. This is why I thought about an alternative method, but maybe it's not doable.
> > > > I will recommend weak acceptance, but lower my confidence a bit. If in the final reviewer discussion it turns out other reviewers easily understand all parts of the paper, I will further adjust my rating upward.

---

> ### Author Response · Authors · 2022-11-14
> **Response Part 2: New Experiments**
>
> **“You Only Found the Misleading Result With One Model”**
>
> You were also concerned that this result is only found for one out of 5 models, which makes you worried this isn’t a general finding.
>
> To clarify, the misleading prompt only _degrades zero-shot accuracy_ substantially for one of the 5 models, but finding prompts that cause large zero-shot accuracy decreases is not one of the main goals of our work. Our method – which is the main contribution of our work – still works well on all the other models in this setting; there is simply not a large zero-shot accuracy gap for our method to close in the first place for other models, relative to the non-misleading setting. We clarified this point in the revised paper.
>
> We explicitly tried to hedge our results for this experiment by emphasizing that it is “an initial proof of concept” and including a footnote with certain caveats. We believe this is a suggestive experiment, but that fully exploring the effects of misleading prompts is outside the scope of our work. If you think our caveats are still insufficient, please let us know.
>
> **Evidence for “CCS Can Find intermediate Representations of Truth”**
>
> Your final main concern seems to be that you’re not sure whether the internal knowledge found by our method is qualitatively de-correlated from the knowledge in the output.
>
> To help address this concern, we restructured 3.3.2 to make our argument clearer. In particular, we ran a couple additional experiments to further illustrate the gap between the knowledge we recover and the model’s outputs.
>
> First, we show that in the misleading prompt setting, it is essential that we use intermediate-layer hidden states rather than the last-layer hidden states. In particular, while we had shown that CCS with the intermediate-layer (encoder) hidden states are robust in this setting (3.2.2), we now also tested CCS in this setting when we use the last-layer (decoder) hidden states, and find that in this case there is a large accuracy drop (81.0% to 73.5%) for CCS – almost as large as the drop in zero-shot accuracy (80.4% to 70.9%). This suggests that compared to the later layers of a model, intermediate layers are more robust and less correlated with the model outputs, which CCS can take advantage of.
>
> Second, we show that CCS still works when model outputs are mostly uninformative. In particular, when we use masked language models without [MASK]ing any input tokens, and when we prompt models so that the labels used to construct contrast pairs appear in the middle of a prompt rather than at the end, model outputs aren’t likely to be meaningful (see the revised Section 3.3.2 and Appendix C for details). We evaluate CCS in this setting, and show that it can still work well (93.7% accuracy for Amazon), and verify that the outputs indeed aren’t very informative (attaining a much lower zero-shot accuracy of 71.6% accuracy in this setting). This provides further evidence that CCS does not just recover knowledge in the model outputs.
>
> We hope these changes address your main concern. If you still think our paper should be rejected because “there are too many doubts around the main claim that the internal knowledge found via the proposed method is actually qualitatively de-correlated from the knowledge in the output”, please let us know what else we can do to clear up this concern.
>
> **Alternative methods**
>
> Thank you for your suggestions for alternative methods to show that internal truth values are present independent of the output.
>
> First, you suggest replacing positive and negative labels with completely uninformative/random text (since we normalize out the effect of “yes” and “no”).
>
> To clarify, our method should not still work in this case. It is true that we remove the effect of the _label_ “yes” or “no” in our normalization step, but the intuition behind our method is that the model’s representations still contain semantic information about whether the given answer is _actually true or false_, in a way that results in opposite truth values for “yes” and “no”. In contrast, with uninformative/random text, the input won’t have a truth value, and certainly won’t have opposite truth values with two different pieces of random text. We tested this empirically in a simple setting (DeBERTa-v2 and IMDB) because of your suggestion, and verified that using uninformative text results in  nearly random accuracy with our method.
>
> You also suggest “don't alter x_i, but train the model with two different classification heads,” but we’re unfortunately not sure what this means exactly. However, we're happy to test this too if you'd like to clarify.
>
> **Conclusion**
>
> We hope our responses address your biggest concerns. Do you still think our paper should be rejected? If so, please let us know what your remaining major concerns are so that we can address them as well. If not, please update your score accordingly.
>
> Thank you very much again for your detailed and thoughtful review!

---

### Official Review · Reviewer_YocT · 2022-10-24

**Confidence:** 4
**Correctness:** 3
**Technical Novelty And Significance:** 3
**Empirical Novelty And Significance:** 3
**Recommendation:** 6

**Clarity, Quality, Novelty And Reproducibility:**

- Clarify/Quality: The paper is very clear and is overall well-written.
- Novelty: The contribution on exploring the internal consistency of language models in an unsupervised way is novel.
- Reproducibility: code and prompts for zero-shot are provided in the paper / supplementary materials.

Minor:
- Section 3.3.1, 2nd paragraph, "transfer wells" -> "transfers well"
- Table 1, what is the difference between "CCS" vs "CCS (All data)"? I thought CCS already used all 1k balanced subsamples of the training set?

**Strength And Weaknesses:**

Strengths:
- To the best of my knowledge, exploring the internal consistency of language models in an unsupervised way is a novel contribution. The authors proposed an interesting method via optimizing the LM with a consistency and a confidence loss to figure out the more likely choice.
- The empirical results are significant across 4 out of 6 models, and across 10 tasks.

Weakness:
- The proposed approach applies only to questions that can be converted with binary answers (i.e., the answer choices are fixed and are from a very limited set, otherwise the inference cost would increase substantially).
1) This largely constrains the applicability of this method to many tasks, e.g., math tasks that have numbers as answers, which would require generating an infinite number of binary-answer questions.
2) for some tasks the conversion might be non-trivial and requires additional human processing.

- Although the proposed method doesn't use any labels, it still uses input examples and requires a fairly large number of training examples to perform well (Figure 5) due to the optimization procedure. This could pose a challenge compared to true zero-shot learning which doesn't require any examples at all. For any new task, how to obtain a large number of training examples (even without labels)?

- Table 1 presents average performance. Is there a more detailed breakdown of performance gains across different tasks? Are there categories of tasks that benefit from the method more vs less?

- Relatedly, as the authors have stated, CCS relies on the assumption that true/false inputs can be separated reasonably well in the activation space. Are there analysis regarding what are tasks that satisfy this property and what tasks are less easily separated?

**Summary Of The Paper:**

This paper proposes Contrast-Consistent Search (CCS), that tries to identify latent knowledge in language models in an unsupervised manner. The process is to convert each question into binary-answer pairs, then train the LM in an unsupervised way via optimizing a consistency loss (probabilities of opposite answers should sum to 1, and a confidence loss to avoid degeneration.

The proposed method outperforms (calibrated) zero-shot accuracy by 4% on average over 10 question-answering tasks. The authors did some additional analysis like CCS's robustness to misleading prompts and the transferability of CCS.

**Summary Of The Review:**

Despite its limited setting (tasks need to be converted to questions with binary answers, and the need of a large number of unsupervised samples), this paper's contribution is quite novel with relatively strong empirical results, thus I recommend weak accept.

---

> ### Author Response · Authors · 2022-11-14
> **Our Approach Is Flexible**
>
> Thank you very much for your careful review. We are glad you liked our paper.
>
> **The approach requires converting questions to binary answers, so it doesn’t work for e.g. math tasks.**
>
> We agree that our approach requires converting questions to binary answers; we focused on this case because we believe it captures the core difficulty of the problem we are focused on while being as simple as possible.
>
> That said, the idea is that this can still work to _verify_ answers to arbitrary open-ended domains (such as math) by applying it to model generations. For example, if a model generates an answer to a math question, we could in principle use our approach to determine whether or not to trust that output.
>
> **For some tasks the conversion of questions to binary answers may require additional human processing**
>
> This is technically true right now, but the human processing it requires is very minimal (it only requires coming up with a new prompt, which is easy to do manually in just a couple minutes), and if you don’t want to create a new prompt you could alternatively use a generic prompt such as “Is the following true or false? &lt;some natural language statement> Answer: &lt;true/false>” that should work for any task.
>
> **The approach requires a fairly large number of training examples to perform well (Figure 5)**
>
> To clarify, our approach _does not_ require a large number of unlabeled training examples to perform well; for example, we show in 3.3.3 that it is often competitive with only, say, 16 examples.
>
> **How do you obtain new examples for a new task?**
>
> Most simply, you generally don’t need new examples for a new task in the first place; our transfer results (3.3.1) show that we can avoid using any task-specific examples by simply training on examples from a different task (e.g. Amazon or DBPedia) then transferring the learned classifier to new tasks; indeed we show in section 3.3.1 that training on the Amazon dataset then transferring to other datasets outperforms CCS without transfer by 0.6%.
>
> If we _did_ want task-specific examples, one option is to have a language model _generate_ examples. Another option is to manually construct a handful of examples: since you don’t need many training examples, and you don’t need answers to questions, this is often easy to do.
>
> **What is the breakdown of performance across different tasks?**
>
> This is shown in the transfer results (the last rows of Figure 2 and Figure 11).
>
> **When can true/false inputs be reasonably well-separated in activation space?**
>
> Intuitively this should be true if the model “knows” the task well, so it will tend to be true of simpler tasks (that current models are better at) and less true of harder tasks (for which current models have more mixed performance). Logistic Regression (LR) accuracy provides a measure of how well-separated a given dataset is in activation space. We show task-specific LR results in the right-most of the four large columns in Figure 11, with no-transfer results in the last row for each model.
>
> **Conclusion.**
>
> We hope our responses clear up any confusions you may have had. Do you still think our method only applies in a limited setting (leading you to giving it a score of only weak accept)? If so, please let us know what else we can do to help address your concerns.
>
> Thank you very much!

---

> > ### Comment · Reviewer_YocT · 2022-11-18
> > **Reply to response**
> >
> > Thanks for the response.
> >
> > > re: applicability of the proposed method to math tasks / any tasks where the answer requires free-text generation.
> >
> > For math tasks, verifying the solution is a useful application but I think it still means the applicability of the method is very limited. For any tasks where the space of the answers is infinite (this applies to other science questions like physics or chemistry, or for any questions where the answer requires free-text generation), in order to "solve" a task, you would have to use another model to generate an answer, then use your method to verify the answer, and if the initial model is not good (even assuming your proposed method is 100% accurate for verification), it could take forever to solve those tasks. It would be interesting to show if the proposed method could be generalized to an answer space that's not binary.
> >
> > > re: The approach requires a fairly large number of training examples to perform well (Figure 5)
> >
> > Thanks for the clarification. What I meant is compared to true zero-shot settings (used as the baseline), your method requires more examples (at least 16, sometimes 32/64) to get a relative good performance. Also, 16 examples can get a somewhat good performance but I think your figure showed the method can benefit from more examples and in some cases only 512 examples can get the performance to a competitive level (so in practice people might want to use more examples to be on the safer side such that the performance converges).
> >
> > > re: obtain new examples and transfer results.
> >
> > It seems like the transfer results vary quite a bit in terms of performance, e.g., the performance of QNLI and PIQA suffers quite a bit during transfer (even trained on themselves). So it is hard to say for a new task, how well it is to transfers from an existing task using existing data. More importantly, which task should be used, a task more similar to the target task, or Amazon/DBPedia in general? I highly doubt Amazon/DBPedia transfers well to every new task, but it also doesn't seem like task similarity plays an important role here based on the results in Figure 2. Maybe a more in-depth study could be done here.
> >
> > I would like to thank the authors for the detailed response, but since most of my concerns remain, I would like to keep my current rating.

---

> ### Author Response · Authors · 2022-11-18
> **Gentle Reminder To Let Us Know If You Have Any Remaining Questions**
>
> Thank you again for your thoughtful and positive review of our paper! We just wanted to check in one last time before the deadline tomorrow: do you have any questions about our responses to your review? If you do, we would be very happy to answer them (as long as we have time). If not, and our responses addressed your main concerns, do you think our paper should now be a clear "accept"? Thank you!

---

### Author Response · Authors · 2022-11-14
**Revised Paper**

We uploaded a revised paper which aims to address several comments:

* We clarified several of our claims in section 3 to clear up a few confusions that reviewers had.
* We added new experiments to further support the claim that our method can find knowledge not available in the outputs, and restructured 3.3.2 to present our evidence for this claim more clearly. In particular:
    * We show that CCS can work in a setting where the outputs are uninformative: specifically, for models trained only with a masked language modeling objective (pretrained DeBERTa-v2 without NLI finetuning), even when we don’t [MASK] any input tokens, and even when we change the prompt so that the positive and negative labels used to construct contrast pairs appear in the _middle_ of a prompt rather than as the very final tokens.
    * We had shown in our paper that when we use the misleading prompt, the performance of CCS remains high even though zero-shot accuracy drops substantially. This is true in the _encoder_ of UnifiedQA; we now additionally verify that when we use the _decoder_ instead, accuracy drops a similar amount to zero-shot accuracy (nearly 10%). This provides additional evidence that, compared to the later layers, intermediate layers can be (1) less correlated with the outputs and (2) more robust – properties that our method takes advantage of. This further suggests that our method does not simply find knowledge contained in the model outputs.
* We added some additional minor implementation details to the appendix related to how we format inputs for encoder-decoder models.
* We made various minor improvements to the writing and formatting of the paper thanks to your suggestions.

---

### Decision · Program_Chairs · 2023-01-20

**Decision:**

Accept: poster

**Justification For Why Not Higher Score:**

As pointed out by the reviewers and also partly agreed by the authors, the proposed method is only applicable to certain inputs and might not be general enough yet. Future work towards its generalization would be nice.

**Justification For Why Not Lower Score:**

Despite the paper's weaknesses mentioned above, the ICLR audience would love to see such a novel contribution in the conference that tackles a new, important problem with sound experimental results.

**Metareview: Summary, Strengths And Weaknesses:**

Overall, the problem that the paper is tackling is a very important problem in the field and the method is novel. The experimental results are also strong. There were some miscommunications and concerns on the limitation of the proposed method, but the former has been mostly resolved via rebuttals and the latter does not seem to outweigh the paper's strengths.


Strengths:
- All reviewers agree that exploring the internal consistency of language models in an unsupervised way is a novel contribution and well motivated.
- Most reviewers agree that the empirical results are also sound.

Weaknesses:
- While the paper was very clearly written in general, there were some crucially unclear parts in the paper, which required back-and-forth communications between the reviewers and the authors to be clarified.
- Some reviewers were concerned that the experimental results might not necessarily support the claims made.
- Some reviewers were concerned that the proposed method is only applicable to certain domains, e.g. questions that can be converted binary answers.

**Note From Pc:**

if the above contains the word "oral" or "spotlight" please see: "oral" presentation means -> notable-top-5% and "spotlight" means -> notable-top-25%. As stated in our emails, we are disassociating presentation type from AC recommendations

**Summary Of Ac-Reviewer Meeting:**

While the paper received a score of 6 from all reviewers, all reviewers agreed that this paper has a niche contribution that many of them would like to see in a conference like ICLR that encourages new perspectives even if they are not yet fully verified.